# Rapid cloning-free mutagenesis of new SARS-CoV-2 variants using a novel reverse genetics platform

**Enja Tatjana Kipfer[1†], David Hauser[1†], Martin J Lett[1], Fabian Otte[1], Lorena Urda[1], Yuepeng Zhang[1], Christopher MR Lang[1], Mohamed Chami[2], Christian Mittelholzer[1], Thomas Klimkait[1]\***

[1]Molecular Virology, Department of Biomedicine, University of Basel, Basel, Switzerland; [2]BioEM Lab, Biozentrum, University of Basel, Mattenstrasse, Basel, Switzerland

**Abstract** Reverse genetic systems enable the engineering of RNA virus genomes and are instrumental in studying RNA virus biology. With the recent outbreak of the coronavirus disease 2019 pandemic, already established methods were challenged by the large genome of severe acute respiratory syndrome coronavirus 2 (SARS-CoV-2). Herein we present an elaborated strategy for the rapid and straightforward rescue of recombinant plus-stranded RNA viruses with high sequence fidelity using the example of SARS-CoV-2. The strategy called CLEVER (CLoning-free and Exchangeable system for Virus Engineering and Rescue) is based on the intracellular recombination of transfected overlapping DNA fragments allowing the direct mutagenesis within the initial PCR-amplification step. Furthermore, by introducing a linker fragment – harboring all heterologous sequences – viral RNA can directly serve as a template for manipulating and rescuing recombinant mutant virus, without any cloning step. Overall, this strategy will facilitate recombinant SARS-CoV-2 rescue and accelerate its manipulation. Using our protocol, newly emerging variants can quickly be engineered to further elucidate their biology. To demonstrate its potential as a reverse genetics platform for plus-stranded RNA viruses, the protocol has been successfully applied for the cloning-free rescue of recombinant Chikungunya and Dengue virus.

**\*For correspondence:**
thomas.klimkait@unibas.ch

[†]These authors contributed equally to this work

**Competing interest:** The authors declare that no competing interests exist.

## eLife assessment

This study describes CLEVER, an improved method for fast and efficient rescue and mutagenesis of SARS-CoV-2. While the principle of this method is not new, this work significantly improves upon existing protocols, providing an **important** advancement in the field of viral infectious clones. **Convincing** proof-of-concept experiments were performed that demonstrate the utility and efficiency of the method.

## Introduction

Severe acute respiratory syndrome coronavirus 2 (SARS-CoV-2) is the causative agent of the human coronavirus disease 2019 (COVID-19), which is responsible for the global pandemic that emerged in November 2019 (*Gorbalenya et al., 2020*; *Wu et al., 2020*; *Zhou et al., 2020*). The virus possesses a positive-strand RNA genome close to 30 kb encoding at least 26 proteins, flanked by 5′ and 3′ untranslated regions (UTRs) and a poly(A) tail at the 3′ terminus (*Lu et al., 2020*).

Principally, the targeted engineering of RNA viruses to study viral biology requires the conversion of the RNA genome into a cDNA copy before it can be manipulated. Based on the large coronavirus

genome size of 27–32 kb, and the presence of typical sequences as nucleotide runs or so-called 'poison sequences' that are hard to amplify in bacteria, it took quite some time before the first full-length coronavirus cDNA clone was published in 2000 (*Almazán et al., 2000*). The initially described bacterial artificial chromosome (BAC) cloning technology was quickly followed by other but similarly laborious techniques to yield infectious cDNA clones of coronaviruses, such as in vitro ligation or vaccinia-based cloning techniques (*Thiel et al., 2001*; *Yount et al., 2000*).

During the emergence of severe acute respiratory syndrome virus (SARS-CoV) in 2003 or Middle East respiratory syndrome virus (MERS-CoV) in 2012, applicable coronavirus reverse genetics methods remained largely unchanged. When SARS-CoV-2 emerged in 2019, previously established coronavirus reverse genetics methods including in vitro ligation and BAC cloning were rapidly adapted to SARS-CoV-2 (*Fahnøe et al., 2022*; *Xie et al., 2020*; *Ye et al., 2020*). Then, additional methods established for other RNA viruses became available: transformation-associated recombination (TAR)-cloning in yeast (*Thi Nhu Thao et al., 2020*) and the 'circular polymerase extension reaction' (CPER) (*Amarilla et al., 2021*; *Torii et al., 2021*). However, TAR-cloning still requires multiple steps of in vitro manipulation, and CPER still relies on the in vitro assembly and subsequent transfection of the 30 kb full-length product. Further, the authors of CPER report the design of unique primers to be critical for efficient in vitro assembly, and optimal PCR settings need to be elaborated. Another DNA-based method has been described by Lamballerie et al. in 2014 for the much smaller flavivirus model of about 10 kb (*Aubry et al., 2014*). This method, termed 'infectious subgenomic amplicons' (ISA), allows transfected overlapping DNA fragments to recombine within the eukaryotic cell into a full-length genome copy. The technique has recently been adapted to SARS-CoV-2 (*Mélade et al., 2022*), eliminating a prior in vitro assembly into a full-length viral cDNA copy or the in vitro transcription and RNA transfection.

Here, we describe an ISA-based method and show its efficient and versatile use with the 30 kb genome of SARS-CoV-2. Largely independent of the inherent limits of the unique size of this virus, we demonstrate a reliable, straightforward, and reproducible reconstitution inside the target cell with high success rates. Extensive next-generation sequencing (NGS) analysis reveals high sequence integrity of the rescued viruses. Furthermore, we demonstrate that recombinant virus is readily rescued from various cell lines or using different transfection methods, making this protocol highly efficient and broadly applicable.

We developed the highly productive genome engineering and reconstitution strategy 'CLEVER' (CLoning-free and Exchangeable system for Virus Engineering and Rescue) as a unique feature. It describes a very efficient technique to rapidly mutagenize or quickly insert defined insertions or deletions into the SARS-CoV-2 genome without any need for time-consuming intermediate cloning steps. In addition, a new protocol enables a direct virus rescue from viral RNA preparations. As proof of concept, we engineered an ΔORF3a (open-reading frame) mutant into an Omicron XBB.1.5 background of SARS-CoV-2 from a clinical isolate within days after the first emergence of this new important virus variant. Notably, the protocol has been adapted and applied for the rescue of recombinant Chikungunya virus (CHIKV) and Dengue virus (DENV) from viral RNA directly, without the need for any further cloning steps.

The CLEVER process therefore provides a platform for broad use in multiple virus applications for various plus-stranded RNA viruses and relevant pathogens far beyond SARS-CoV-2. The technical design and process reflect a formidable and powerful strategy that broadens the general molecular applicability and pushes the limits of effective reverse genetics. The platform is highly user-friendly for multiple new applications and various viruses of interest or newly emerging, unknown pathogens in research and/or of clinical importance.

## Results

### Optimized DNA-based recovery of authentic full-length SARS-CoV-2 virus

While most previously published techniques for the recovery of full-length, infectious coronaviruses require several intermediate steps (*Kurhade et al., 2023*), we found the DNA-based method by Mélade and colleagues an attractive basis for our work. It introduces overlapping subgenomic DNA fragments, covering the entire virus genome, into permissive cells. DNA recombination by the cellular machinery then leads to the generation of a full-length viral genomic cDNA copy from which plus-strand RNA is

transcribed, starting a complete viral replication cycle. Accordingly, we inserted a heterologous promoter as well as critical regulatory elements 5′ and 3′ of the viral genome (*Figure 1A*). Eight overlapping fragments spanning the whole SARS-CoV-2 genome were PCR amplified and transfected in equimolar ratios into HEK293T cells. The resulting propagation of recombinant SARS-CoV-2 (rCoV2) was assessed by the developing cytopathic effect (CPE) once culture supernatant was passaged onto permissive Vero E6 cells (*Supplementary file 1*). A first CPE was typically observed after 6–8 days post-transfection (dpt). Whereas co-transfection of nucleocapsid (N) mRNA or N-expressing plasmid has been reported to be critical for a successful virus recovery (*Xie et al., 2020*), we found no such improvement, as it has been reported previously for a different DNA-based reverse genetics method (*Torii et al., 2021*).

Thus, our protocol is exclusively dependent on a single DNA transfection step, and the omission of any mRNA or DNA co-transfection step renders the rescue procedure significantly less laborious. Furthermore, the skipping of external N allowed us to use standard commercial rapid antigen tests for SARS-CoV-2 to semi-quantitatively monitor a successful virus rescue within minutes. Moreover, this test is done inside the safety facility, eliminating the cumbersome sample export into a standard lab for RT-PCR and viral genome determination (*Figure 1C*).

In a first optimization step, we reduced the number of necessary DNA segments from eight to four. This simplified the PCR protocol and reduced the needed intracellular recombination events from seven to only three for reconstituting a full-length viral genome.

To test our hypothesis that fewer recombination sites will improve reconstitution efficiency and therefore virus production, the number of HEK293T cells releasing infectious virus after transfection was evaluated. HEK293T cells were either transfected with eight or four fragments and subsequently seeded in 96-well plates at 5000 cells/well (*Figure 1—figure supplement 1A*). On day 7, supernatant from each well was separately transferred onto Vero E6 cells. Emerging infectious virus was assessed by plaque formation. The transfection of four fragments resulted in about one virus-producing cell per 11,000 transfected cells compared to ~1/160,000 for eight genome fragments (*Figure 1—figure supplement 1B*).

To confirm that transfecting only four fragments does not negatively affect the reconstitution fidelity, we compared the replication competence of recombinant viruses to the parental wild-type virus isolate. At the time of the earliest appearance of cytopathic changes (*Figure 1B*), cultures were assessed side-by-side. Plaques of recombinant SARS-CoV-2 reconstituted from four fragments (rCoV2-4fr) were similar in size and shape compared to the wild-type reference (*Figure 1B*). When comparing intracellular viral N protein expression with immunocytochemistry (ICC), staining intensity and patterns correlated very well with the wild-type phenotype (*Figure 1D*). To compare the replication kinetics of recombinant virus and parental SARS-CoV-2, Vero E6 cells were infected at a multiplicity of infection (MOI) of 0.01 with either virus and infectious titers were determined 12, 24, 48, and 72 hr post-infection (hpi) using a plaque assay (*Figure 1E*). Although the recombinant virus showed decreased titers within the first 24 hr, similar titers were reached after prolonged infection as it has been observed previously (*Torii et al., 2021*; *Xie et al., 2020*).

To further compare virion integrity, viral particles were analyzed by transmission electron microscope (TEM). Regarding size, shape, lumen density, and S protein abundance, the in vitro-generated virus was indistinguishable from wild-type virus (*Figure 1F*).

Concluding these findings, the viruses reconstituted in vitro from four fragments show similar phenotypes compared to the parental strain, as previously described for recombinant SARS-CoV-2 (*Amarilla et al., 2021*; *Mélade et al., 2022*; *Thi Nhu Thao et al., 2020*; *Torii et al., 2021*; *Xie et al., 2020*).

The reconstitution process is highly reproducible, and rescue of recombinant SARS-CoV-2 from transfection was successful in different cell lines including HEK293T, HEK293, CHO, and BHK-21 cells (*Supplementary file 1*). In these cases, due to the absence of the ACE-2 receptor, a productive viral propagation depended on co-cultivation with Vero E6 cells. However, virus was directly rescued from ACE2-expressing A549 cells that did not need co-cultivation. In addition to the broad applicability to various cell types, various transfection protocols were successfully tested on HEK293T including electroporation or three different chemical reagents (*Supplementary file 1*).

## Highly faithful genome amplification protocol

Along with the reduction in the number of DNA fragments, we invested in the optimization of the PCR-based genome amplification steps. To establish a stringent protocol, we attempted to minimize the

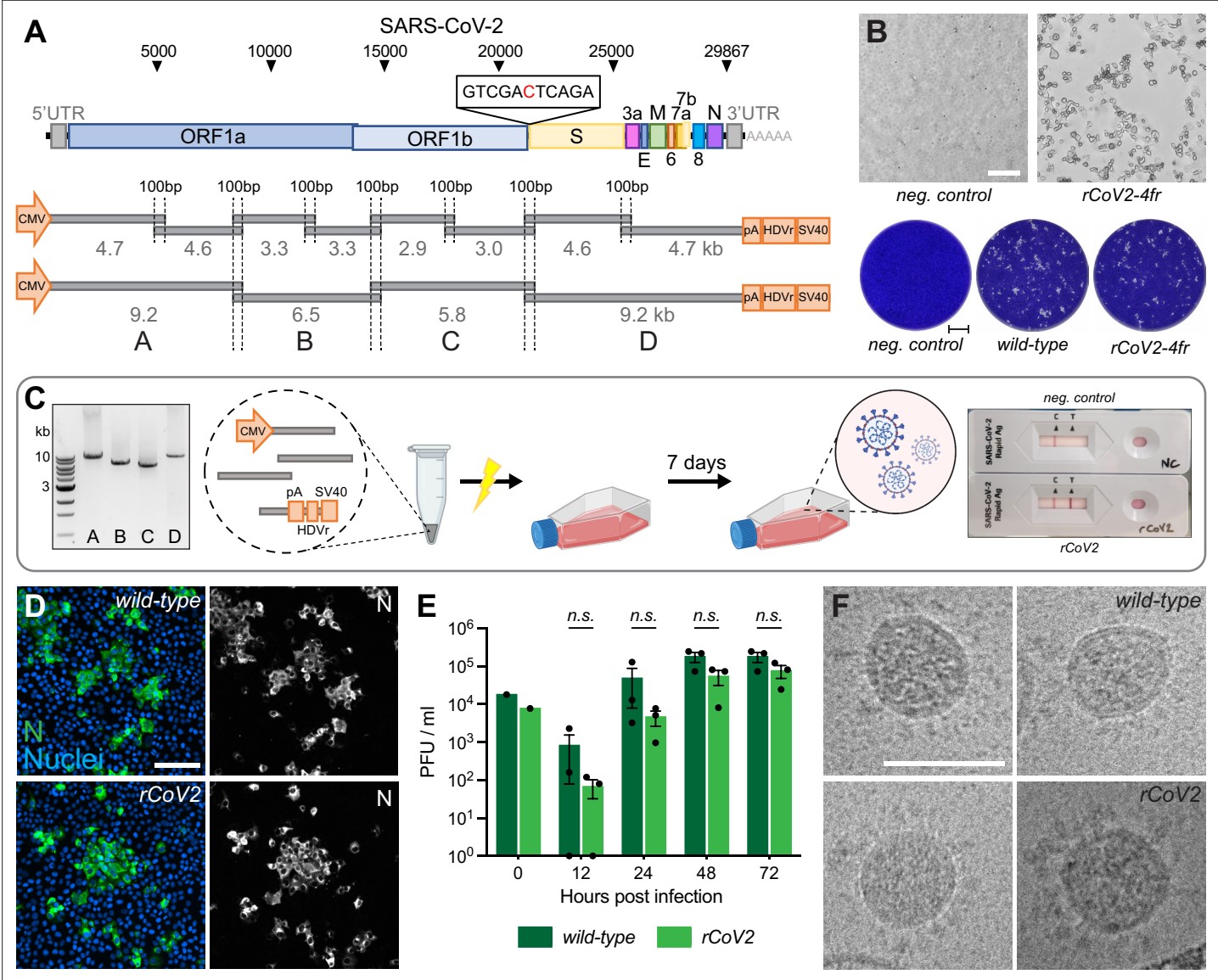

**Figure 1.** Rescue and characterization of recombinant SARS-CoV-2. (**A**) Schematic representation of the SARS-CoV-2 genome and the infectious subgenomic amplicons (ISA)-based method for virus recovery. Eight respectively four overlapping fragments covering the whole SARS-CoV-2 genome were PCR amplified. The heterologous CMV promoter was cloned upstream of the 5' untranslated region (UTR) and a poly(A) tail, HDV ribozyme, and SV40 termination signal downstream of the 3' UTR. (**B**) Infectious virus reconstituted from four fragments (rCoV2-4fr) assessed by cytopathic effect (CPE, top) on susceptible Vero E6 cells by supernatant transfer. Plaque size was compared by standard plaque assay 2 d after inoculation on Vero E6 cells (bottom). (**C**) Workflow for the rescue of recombinant SARS-CoV-2. Four fragments were PCR amplified, mixed in equimolar ratios, transfected into HEK293T cells, and infectious virus was recovered 7 d post-transfection. Commercially available SARS-CoV-2 rapid antigen tests can be used for a quick non-quantitative analysis. (**D**) Detection of intracellular SARS-CoV-2 nucleocapsid protein (N, green) and nuclei (Hoechst, blue) in Vero E6 cells infected with parental wild-type or recombinant virus by immunocytochemistry. (**E**) Growth kinetics of recombinant virus and its parental wild-type virus. Vero E6 cells were infected in triplicates at a multiplicity of infection (MOI) of 0.01, supernatant was collected 12, 24, 48, and 72 hr post-infection and analyzed by plaque assay. Cell layers were washed 2 hr post-infection. Data represents mean ± SEM, analyzed with multiple *t*-tests and Benjamini, Krieger, and Yekutieli correction (N = 3 individual biological replicates, n = 3 technical replicates). (**F**) Cryo-transmission electron microscope pictures of parental wild-type virus and recombinant virus in glutaraldehyde-fixed samples. Scale bar is 100 µm (top) and 2 mm (bottom) in (**B**), 20 µm in (**D**), and 100 nm in (**F**).

The online version of this article includes the following source data and figure supplement(s) for figure 1:

**Source data 1.** Uncropped agarose gel image of *Figure 1C*.

**Figure supplement 1.** Clonal virus populations and reconstitution efficiency.

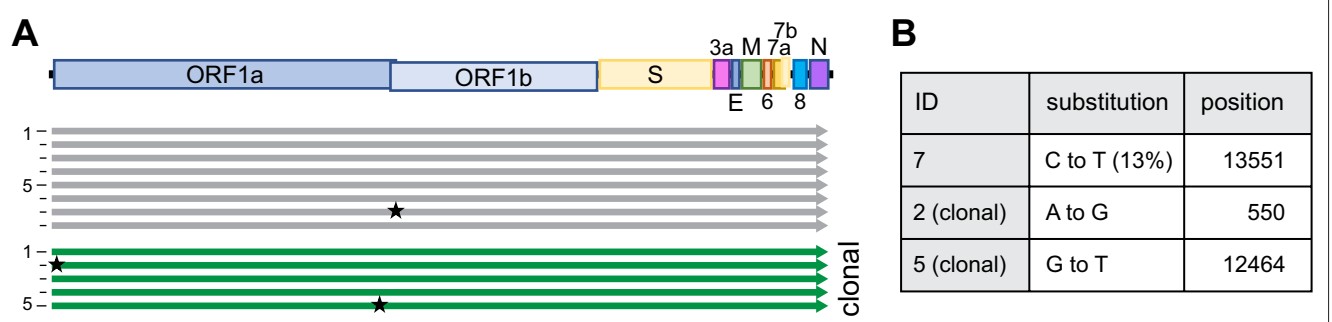

**Figure 2.** Sequence integrity using CLEVER. (**A**) Schematic representation of the sequence alignment of recombinant viruses sequenced by next-generation sequencing (NGS), mutations with a relative abundance of >10% are indicated with a star. A total of eight bulk (gray) and five clonal (green) populations were analyzed. (**B**) Details on substitution and position in the genome. For a more detailed analysis see *Supplementary file 2*.

inherent PCR-based errors in the product by applying certain rules: use of a high-fidelity polymerase, limiting the number of amplification cycles to 25, use of a high-template input (20–40 ng/25 µL reaction), and the pooling of at least eight parallel PCR reactions.

To analyze the sequence integrity, eight independently rescued viruses were subjected to NGS and mutations with a relative abundance of >10% in the entire virus population were analyzed (*Figure 2*, *Supplementary file 2*). To minimize any possible carryover of input DNA, virus was passaged twice 1:1000 on Vero E6-TMPRSS2 cells before viral RNA was extracted.

Of note, we strictly used TMPRSS2-expressing cells for any propagation of SARS-CoV-2 as the expression of TMPRSS2 has been shown to effectively prevent the reported loss of the S1/S2 cleavage site (*Sasaki et al., 2021*). In position 20,949 of the viral genome, we deliberately introduced a silent mutation that serves as a genetic marker (T20949C, introducing a Sal I restriction site) to discriminate our recombinant viruses from any accidental contamination with a clinical isolate (*Figure 1A*). This genetic marker was confirmed in all reconstituted viruses.

Using the above-mentioned stringent protocol, among eight reconstituted recombinant viral genomes, only one minor single-nucleotide polymorphism (SNP) was identified overall (*Figure 2B*). None of the SNPs occurred at the S1/S2 cleavage site, which others reported to be prone to spontaneous changes in vitro (*Sasaki et al., 2021*).

In any given culture, several cells within the same well might have the ability to simultaneously initiate the production of infectious virus after transfection (compare *Figure 1—figure supplement 1B*), and any minority of an emerging mutated virus genome might be missed in the subsequent bulk sequencing of a heterologous population. For analysis, we thus sequenced clonal virus populations initially produced from one successfully transfected HEK293T cell. Reconstituted from four fragments, 174/384 wells (45%) turned virus-positive (*Figure 1—figure supplement 1B*), and 18 clonal viruses (thus representing more than 10% of all recovered viruses) were subjected to Sanger sequencing of the entire S gene. Not a single SNP was detected. To further confirm the observed sequence fidelity on the clonal level, NGS data of five clonal virus populations was analyzed: overall, six SNPs were detected, with five SNPs found within the same virus. Further, among these five SNPs, four were found within the 5′ UTR. Also, 3/5 clonal populations had 100% sequence integrity, ultimately proving the capacity of high-fidelity DNA polymerases (*Figure 2*, *Supplementary file 2*).

## Highly versatile and rapid inter-genomic gene recombination

A unique property of the ISA-based method resides in the fact that the DNA fragments do not need to be assembled before transfection, but that the choice of a fragment, for example, representing a special virus isolate or harboring a reporter sequence, remains exchangeable until the very last step before transfection. For this study, fragments were PCR amplified from different sources such as plasmid DNA or viral RNA in clinical specimens or even de novo synthesized linear dsDNA was used. Herein, it appeared important to test whether the DNA fragment size was limiting or whether certain sizes are preferred for the process. We successfully utilized fragment sizes from as large as 9 kb to only 500 bp (*Supplementary file 1*). For these tests and the targeted introduction of specific mutations, different genomic regions were tried as recombination sites. Importantly, recombination sites were

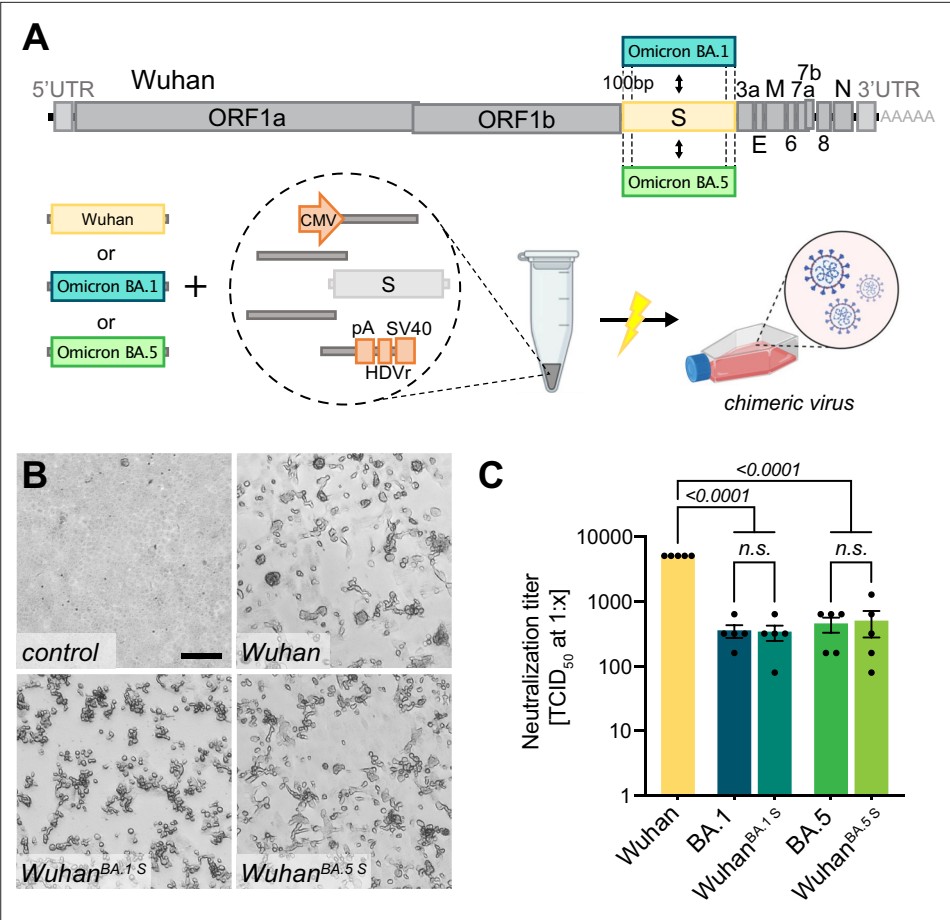

**Figure 3.** Creating chimeric virus by fragment exchange. (**A**) Schematic representation of the exchange of individual fragments. Shown is the replacement of the Wuhan S sequence by the sequence encoding for the Omicron BA.1 or BA.5 S gene. The genetic background (outside of S) is kept in the original Wuhan sequence. All fragments needed to reconstitute the virus were transfected and chimeric virus was rescued. (**B**) Successful rescue of infectious chimeric virus was assessed by cytopathic effect (CPE) formation on Vero E6 cells. Scale bar represents 100 μm. (**C**) Titers of neutralizing antibodies against different SARS-CoV-2 S gene variants were validated in sera from vaccinated individuals. Sera were incubated with parental wild-type virus (Wuhan), Omicron BA.1 or BA.5 clinical isolates (BA.1, BA.5), as well as chimeric viruses having the Wuhan background combined with either the Omicron BA.1 S gene (Wuhan$^{BA.1\,S}$) or Omicron BA.5 S gene (Wuhan$^{BA.5\,S}$). Neutralizing titers were determined with a neutralization assay and TCID$_{50}$ read-out. Data represents mean ± SEM, analyzed with one-way ANOVA followed by Bonferroni's test (N = 5).

chosen independently of GC content or the presence of repetitive sequences (**Supplementary file 3**). Irrespective of the manipulated region of the viral genome, all in silico designed recombination sites successfully yielded infectious viruses. This allows for very high freedom in choosing sites for homologous recombination across the entire genome as long as a 100 bp homology region is kept to the neighboring fragment (shorter overlaps have been used but were not further investigated in this study).

We demonstrate the high flexibility of the system by exchanging the region that is encoding for the SARS-CoV-2 S gene: we introduced the S gene sequence of several newly arising variants including Omicron BA.1 or BA.5 while keeping the background sequence of the original strain (herein referred to as Wuhan) in the rest of the genome (**Figure 3A**). The target sequence of interest was either amplified from commercial and in-house cloned plasmids or directly from clinical isolates without further cloning. The chimeric virus was rescued (**Figure 3B**) and confirmed by sequencing specific regions to discriminate variants (**Supplementary file 4**). To further determine the functionality and integrity of the newly introduced S gene, recovered viruses were tested in a virus neutralization assay

(*Figure 3C*): the chimeric viruses Wuhan[BA.1 S] and Wuhan[BA.5 S] or clinical isolates of Wuhan, Omicron BA.1, or Omicron BA.5 were subjected to neutralization by dilutions of human sera from vaccinated persons. Not surprisingly, the highest neutralizing titers were observed against the Wuhan virus as serum samples were collected before the introduction of bivalent vaccines. Titers were significantly lower against later virus variants such as Omicron.

Interestingly, the neutralizing titers against chimeric viruses were similar to those of the respective full-length spike homologs, and significantly lower than for the Wuhan virus, indicating that the S gene sequence mostly determines neutralization titers in vitro.

## One-step introduction of point mutations, modifications, or specific gene deletions

As fragments are amplified by PCR, the amplification step can be used for direct manipulations, and primers can be specifically designed to introduce mutations within the homology region without any need for cloning or de novo synthesis of a whole fragment (*Figure 4A*). The CLEVER primer design ensures a 100 bp sequence overlap between the generated PCR products that are either reached by separating the primer annealing sites or by adding a nucleotide stretch to the 5′ end of a primer annealing site to generate the desired 100 bp homology (*Figure 4—figure supplement 1*).

To demonstrate the ease of introducing mutations using CLEVER, an oligonucleotide pair was designed to introduce the widely discussed N501Y mutation responsible for higher transmissibility and infectivity of SARS-CoV-2 (*Liu et al., 2022*). In a second approach, G614 was mutated back to the less favorable amino acid D614 (*Korber et al., 2020*; *Plante et al., 2021*). For both approaches, the two fragments harboring the SNP in the joint overlap sequence were co-transfected with the other fragments needed to complete the whole cDNA copy. The successful introduction of the desired SNP into the rescued virus was confirmed by sequencing (*Figure 4B*, *Supplementary files 1 and 4*). Sequence stability of both the fitness-enhancing but also fitness-impairing mutation was confirmed by resequencing after five passages (*Supplementary file 4*). This demonstrates that specific SNPs can be readily introduced by using primer-specific mutagenesis and that both, beneficial and disadvantageous mutations, can be stably integrated.

Another key application of the versatile CLEVER system is to generate larger genome changes such as the deletion of entire genes or the site-specific introduction of extended sequences within the initial PCR step. To this end, we used the design of oligonucleotide primers that anneal upstream and downstream of ORF3a, 'bridging' the genomic regions 5′ of the gene directly to the genomic region 3′ of ORF3a (*Figure 4A*). With this, we were able to delete the entire gene in one step from the otherwise intact genome. The successful and precise deletion of ORF3a was confirmed by Sanger sequencing (*Supplementary files 1 and 4*) and on the protein level by immunoblot (*Figure 4C*) and ICC (*Figure 4D*).

Moreover, we attempted to site-specifically insert short foreign sequences into the SARS-CoV-2 genome. Due to its short size (66 bp) and easy detection methods, we chose to add a triple FLAG-tag (*Einhauer and Jungbauer, 2001*) to the carboxy-terminus of ORF8 and separated it by a short flexible amino acid linker (GGGGS). Virus that carries the introduced FLAG sequence was successfully rescued (*Supplementary files 1 and 4*), and FLAG expression was confirmed by immunoblot (*Figure 4C*) and ICC (*Figure 4E*). In this example, the overlap sequence (84 bp) needed for recombination completely covers the sequence coding for the flexible linker and 3xFLAG. Consequently, this would allow the one-step integration of the FLAG sequence into any viral gene.

This strategy allowed us to rescue mutant virus within 2 weeks from the initial in silico primer design until virus rescue in Vero E6 cells. The process included primer design and synthesis (4 d), PCR and transfection (1 d), and obtaining a SARS-CoV-2-induced CPE (7 d) (*Figure 4A*).

## Intracellular circularization strategy for a completely cloning-free rescue

Due to the continuing emergence of new variants, fast adaptation of the laboratory strains and mutants/reporters is a constant process. Cloning or de novo synthesizing the complete genome of new variants is laborious and time-consuming. A small modification in our design led to a system for the rescue of recombinant virus from a clinical isolate directly.

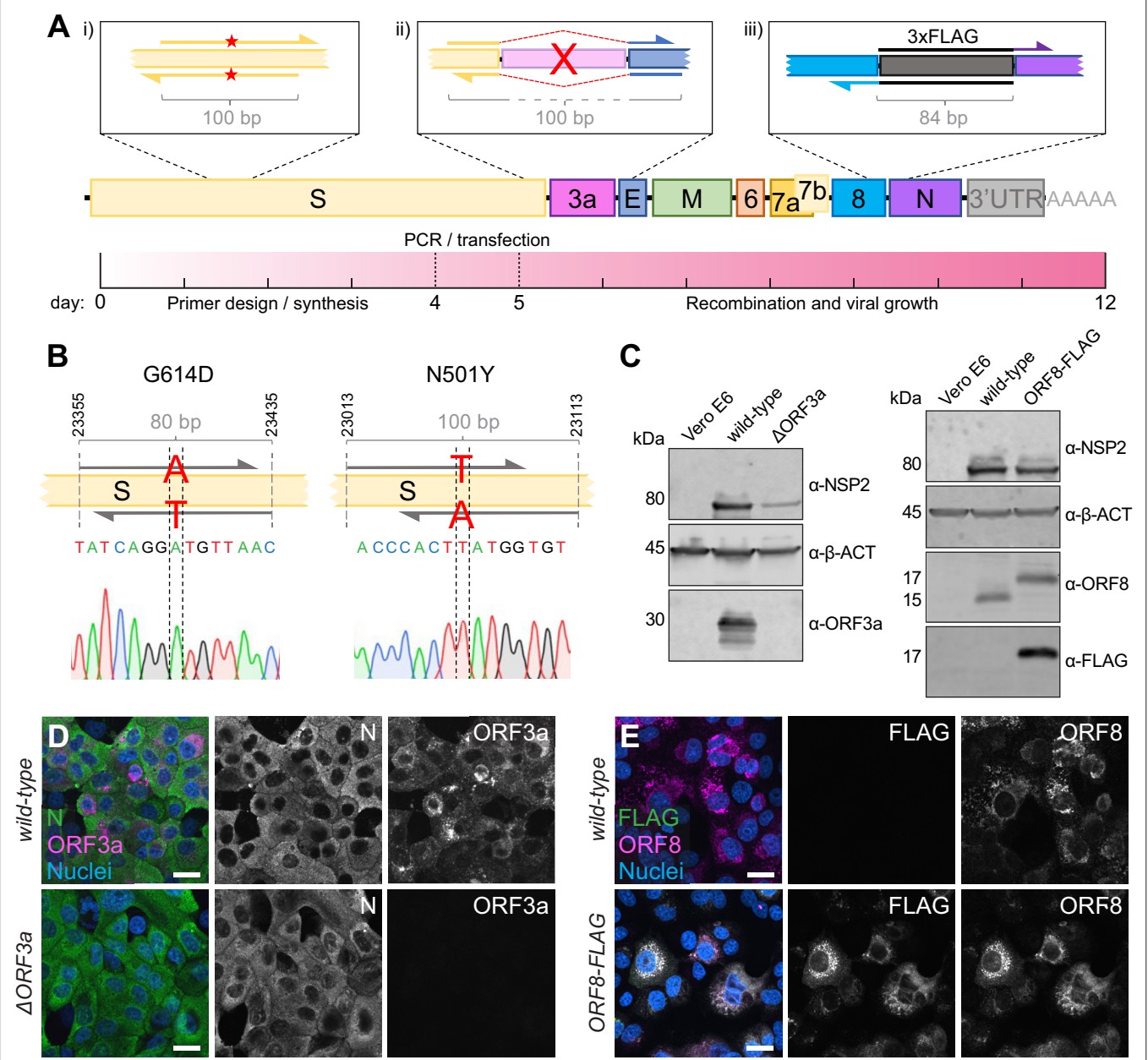

**Figure 4.** Direct mutagenesis using the CLEVER primer design. (**A**) Schematic representation of the CLEVER primer design for direct mutagenesis. Shown is the (i) introduction of small nucleotide changes, (ii) the deletion of larger sequences, here shown for ORF3a, and (iii) the insertion of nucleotide stretches such as 3xFLAG as well as a timeline showing the expected work-flow needed from in silico design to virus rescue. (**B**) Details on the G614D and N501Y substitution within the S gene. Shown is position, primer design, and the integration into the viral genome confirmed by Sanger sequencing. (**C**) Validation of mutations by immunoblot. Shown is the validation of the ΔORF3a (left) and ORF8-3xFLAG virus (right). Vero E6 cells were assessed with α-β-actin (α-β-ACT) and viral infection was detected using α-NSP2. ORF3a expression or ORF8/FLAG expression, respectively, was compared to wild-type infected cells and uninfected controls. (**D**) Validation of ΔORF3a by immunocytochemistry. ΔORF3a virus created by direct mutagenesis was compared to its parental wild-type virus. Shown is the expression of ORF3a (magenta) in both viruses. Nucleocapsid (N, green) expression was used to assess viral infection, nuclei were stained with Hoechst (blue). (**E**) Validation of ORF8-3xFLAG by immunocytochemistry. C-terminal tagging of ORF8 with 3xFLAG was achieved with direct mutagenesis. Shown is the expression of ORF8 (magenta) and FLAG (green) in both viruses. Nuclei were stained with Hoechst (blue). Scale bar is 20 µm in (**D, E**).

The online version of this article includes the following source data and figure supplement(s) for figure 4:

**Source data 1.** Uncropped western blot images of *Figure 4C*.

**Figure supplement 1.** CLEVER primer design for direct mutagenesis.

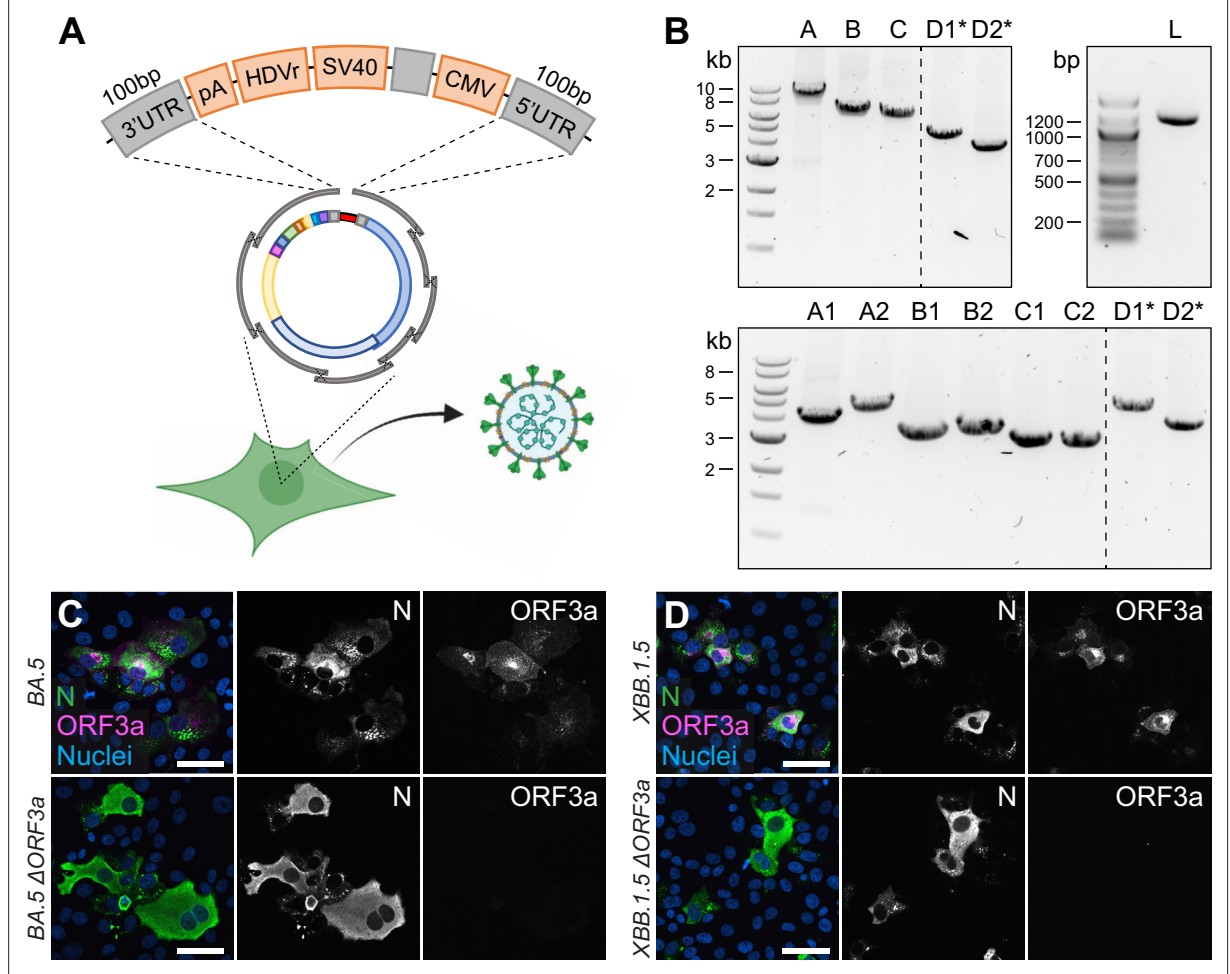

**Figure 5.** Direct rescue and mutagenesis of clinical isolates. (**A**) Schematic representation of the circular assembly within the eukaryotic cell with the linker fragment. The heterologous elements needed downstream of the 3' untranslated region (UTR) (pA, HDVr, SV40) and upstream of the 5' UTR (CMV) are assembled in one fragment, separated by a spacer sequence and flanked by homologous regions needed for intracellular recombination. (**B**) Representative agarose gel pictures from PCR fragments amplified by one-step RT-PCR from viral RNA and the linker fragment (L). Recombinant virus was rescued from five (top) or eight fragments (bottom), plus the linker fragment. Asterisks mark fragments harboring the introduced changes within their homology region. (**C, D**) Validation of (**C**) Omicron BA.5 ΔORF3a and (**D**) XBB.1.5 ΔORF3a by immunocytochemistry. The expression of ORF3a (magenta) in Omicron BA.5 and XBB.1.5 clinical isolates and recombinant ΔORF3a viruses is shown. Nucleocapsid (N, green) expression was used to assess viral infection, nuclei were stained with Hoechst (blue). Scale bar represents 20 μm.

The online version of this article includes the following source data and figure supplement(s) for figure 5:

**Source data 1.** Uncropped agarose gel images of *Figure 5B*.

**Figure supplement 1.** Cloning-free rescue of chimeric virus.

**Figure supplement 1—source data 1.** Uncropped agarose gel images of *Figure 5—figure supplement 1A*.

**Figure supplement 2.** Cloning-free rescue of Chikungunya virus (CHIKV) and Dengue virus (DENV).

**Figure supplement 2—source data 1.** Uncropped agarose gel images of *Figure 5—figure supplement 2B and E*.

Initially, we cloned the eukaryotic expression elements needed for DNA-dependent RNA transcription 5′ and 3′ of the viral genome or purchased complete custom-designed plasmids that included those elements. Others had reported successful viral rescue from viral RNA using the ISA method, but an additional fusion PCR step was essential to incorporate the expression elements described above (*Aubry et al., 2015*).

An additional hallmark of the CLEVER platform describes the rescue of recombinant virus without any need for bacterial cloning, additional PCR steps, or in vitro assembly, resulting in a protocol with the shortest hands-on time reported so far. For this purpose, all expression elements were jointly

assembled into one single 'linker-DNA element,' which contains the 3′ elements (poly(A) tail, HDV ribozyme, SV40 termination signal) but also the CMV promoter (*Figure 5A*; *Amarilla et al., 2021*; *Torii et al., 2021*). At the 5′ and 3′ termini of this linking fragment, 100 bp of the viral 3′ and 5′ UTR, respectively, are added for successful intracellular recombination with the viral DNA fragments. This design recombines intracellular into a circular DNA product, positions the CMV promoter upstream of the 5′ UTR of the viral genome, and places the termination signal downstream from the viral 3′ UTR.

Overall, the entire linker fragment is approximately 1.1 kb in length and leaves the residual fragments needed for genome reconstitution free of any non-viral sequences, meaning that subgenomic fragments can be amplified from viral RNA directly.

We tested this design in the following way: the eight fragments spanning the SARS-CoV-2 genome were directly amplified from viral RNA using a one-step RT-PCR master mix. The products were co-transfected together with the ready-to-use linker fragment, carrying the SARS-CoV-2 overlaps at the termini. Although the number of recombination sites increased from 7 to 9 to create a circular product, viable virus was rescued on day 7 post-transfection (*Supplementary file 1*).

The new CLEVER strategy allowed us to rescue various chimeric viruses with no bacterial cloning step: the genomes of clinical Wuhan, Omicron BA.1, and Omicron BA.5 isolates were directly amplified by RT-PCR (*Figure 5—figure supplement 1A*) and the fragment harboring the S sequence was exchanged before transfection with the corresponding fragment carrying a different S gene variant (*Figure 5—figure supplement 1B*). As a result, replication-competent chimeric viruses were produced that carry a heterologous S gene (*Figure 5—figure supplement 1C*). To confirm the chimeric nature of the rescued viruses, regions of the S gene, as well as the M gene, were Sanger sequenced to discriminate variants (*Figure 5—figure supplement 1D*, *Supplementary file 4*).

In a final step, we combined the two above-mentioned features of the CLEVER system: direct mutagenesis by primer design and direct rescue of recombinant virus from clinical isolates. By doing so, we rescued recombinant virus from the Omicron BA.5 isolate by doing a one-step RT-PCR on extracted viral RNA, but included the primer pair introducing an ORF3a deletion (*Figure 5B*, top). With the emergence of the Omicron XBB.1.5 variant, recombinant ΔORF3a virus was rescued accordingly, but starting from eight fragments instead (*Figure 5B*, bottom). Regardless of the number of fragments transfected, infectious virus was rescued and passaged on Vero E6-TMPRSS2 and introduced changes were confirmed by Sanger sequencing (*Supplementary file 4*). Further, viral N expression (demonstrating infectivity) and ORF3a expression were assessed in ICC (*Figure 5C and D*). In only one step, we were able to create a deletion mutant of two newly emerging variants, and no Omicron BA.5 or XBB.1.5 sequence had to be cloned or de novo synthesized. This eventually demonstrates the wide applicability of the CLEVER platform and proves its rapidity in generating new SARS-CoV-2 mutant variants.

## CLEVER protocol applied to CHIKV and DENV

Although the benefits of the CLEVER protocol mostly come into effect when working on large genomes such as SARS-CoV-2, we demonstrate its applicability to other plus-stranded RNA viruses. Whereas recombinant CHIKV (rCHIKV) has been previously rescued from in-house cloned plasmids using the ISA method, we used the linker fragment to demonstrate the fast rescue of infectious CHIKV directly from extracted viral RNA rather than from cloned plasmids. The primers at the junctions of the linker fragment and the viral genome have been adapted to introduce ~100 bp stretches of homology (*Supplementary file 3*). Further, to distinguish between wild-type and recombinant CHIKV, a silent SNP has been introduced at an inter-genomic recombination site (*Figure 5—figure supplement 2A*). Control experiments revealed that the sole transfection of total RNA results in the rescue of infectious CHIKV or DENV (has not been observed for SARS-CoV-2); therefore, an additional RNase A digestion step following PCR amplification was added. After transfection of all PCR-amplified fragments into BHK-21 cells, a CPE appeared 2 dpt (*Figure 5—figure supplement 2B and C*), and the presence of the silent SNP after Sanger sequencing confirmed the successful use of CLEVER on CHIKV (*Supplementary file 4*).

In a second approach, two patient samples positive for DENV were obtained. The two samples have been identified as belonging to serotypes 1 and 3 (DENV1 and DENV3, respectively) (*Tandel et al., 2022*). Primers have been designed based on published sequences and previous studies (*Siridechadilok et al., 2013*; *Tamura et al., 2022*). For DENV1, two different recombination sites (A and

B) were tested. For all three approaches (DENV1-A, DENV1-B, and DENV3), infectious virus has been rescued and the silent SNP at each recombination site confirmed (*Figure 5—figure supplement 2D–2F*, *Supplementary file 4*). Notably, the same linker plasmid used for SARS-CoV-2 has been used for CHIKV and DENV, and the virus-specific homology region has been introduced by primers.

In summary, we are presenting an optimized, straightforward, and highly versatile cloning and expression system for SARS-CoV-2 and beyond. The new CLEVER system allows us to completely omit tedious in vitro RNA-expression steps and DNA-cloning steps often required for manipulating and reconstituting infectious viral genomes.

## Discussion

The CLEVER strategy describes a highly refined extension of DNA-based methods (such as ISA) and enables a highly versatile use and broad application for the fast rescue and/or mutagenesis of SARS-CoV-2 with no cloning intermediates or procaryotic vectors needed. As examples of its flexibility, we used the specific introduction of the N501Y or G614D point mutations into the S gene, the deletion of ORF3a, or the addition of a completely virus-unrelated FLAG tag to ORF8. Further, by product circularization via a linker fragment, a deletion of ORF3a was directly introduced, and a functional, recombinant virus was rescued in one step starting from viral RNA.

The direct intracellular recombination and viral reconstitution within the eukaryotic cell as utilized by CLEVER provide an enormous benefit for large viral genomes such as SARS-CoV-2 when compared to other reverse genetic methods: (i) the 30 kb genome does not need to be assembled in vitro and (ii) only small DNA fragments are transfected into the producer cell, which goes hand in hand with a much higher transfection efficiency. With CLEVER, in vitro assembly steps and intermediate steps required for most other reverse genetic systems can be skipped (*Figure 6*). Furthermore, individual pieces of DNA can be freely exchanged, and mutations can be introduced just one step prior to transfection with no need for additional cloning. Conveniently, only newly generated plasmids are needed to be sequence verified, whereas the residual fragments remain the same. We demonstrate this feature by rapidly exchanging and analyzing the S gene properties of newly emerging variants. Neutralizing titers correlated with the introduced S gene variant rather than the genomic background. Thus, we present a tool to rapidly study Spike properties with chimeric viruses in vitro, sparing the cloning of the whole 30 kb genome.

We initially amplified 8 fragments as previous studies divided the SARS-CoV-2 genome into 4–12 fragments prior to re-assembly (*Amarilla et al., 2021*; *Herrmann et al., 2021*; *Mélade et al., 2022*; *Rihn et al., 2021*; *Thi Nhu Thao et al., 2020*; *Torii et al., 2021*; *Xie et al., 2021*; *Xie et al., 2020*; *Ye et al., 2020*; *Ye et al., 2020*). The protocol was then simplified by reducing fragment number, enabling high recovery efficiency and thus leading to a library of over 150 viruses, including more than 40 characterized mutant SARS-CoV-2 viruses.

While the initial ISA attempts for SARS-CoV-2 were limited to BHK-21 cells co-cultured with Vero E6 (*Mélade et al., 2022*), we demonstrate successful virus rescue in HEK293T, HEK293, CHO, BHK-21 and A549-ACE2 cells or by using different transfection methods. Thus, we believe that the CLEVER platform can be rapidly established in any laboratory currently doing reverse genetics on SARS-CoV-2 using cell lines and transfection methods of choice.

Another benefit is the freedom of choosing recombination sites as intracellular recombination depends on the length of the homology region rather than GC content. We did not encounter any inefficient overlaps within this study in contrast to the described CPER method where primer design seemed to be a critical step for the assembly of a full-length SARS-CoV-2 genome (*Amarilla et al., 2021*; *Torii et al., 2021*). For CPER, the repetitive occurrence of transcriptional regulatory sequences (TRS) common in coronaviruses can therefore limit the choice of creating new fragment overlaps. Further, unoptimized reaction conditions can favor wrong assemblies. For intracellular recombination like ISA, only correct assemblies will result in an infectious virus and are amplified with repeated infection cycles. Thus, the system is selective for infectious virus.

The key property of the successful CLEVER protocol is the minimization of any PCR-introduced mutations. Up to date, the ISA method has not yet gained much attention in the field of reverse genetics methods for SARS-CoV-2 (*Kurhade et al., 2023*; *Mittelholzer and Klimkait, 2022*; *Wang et al., 2022*), despite its straightforward protocol. PCR amplification is still highly associated with polymerase-introduced mutations, and the PCR amplification step has been described as an error-prone

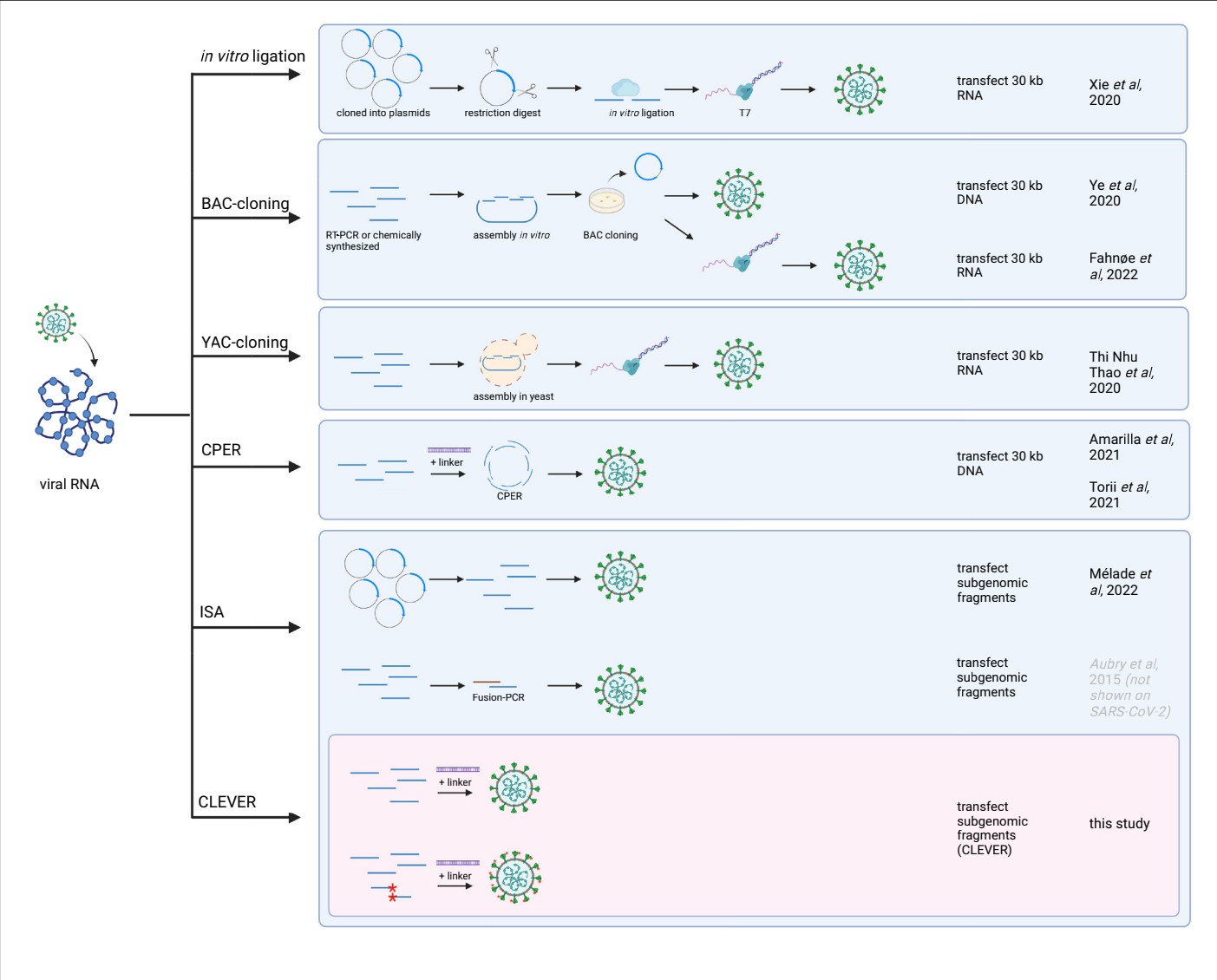

**Figure 6.** Overview of reverse genetics methods for SARS-CoV-2. The most commonly used reverse genetics systems for the rescue of recombinant SARS-CoV-2 are listed and the prominent intermediate steps are depicted. Note that this is a schematic summary and additional steps (such as purification, linearization before transcription, etc.) or small aberrations of the protocol (e.g., different starting material) can apply. The first groups reporting the successful adaptation to SARS-CoV-2 are mentioned. For the CLEVER method, additionally the direct mutagenesis within the initial RT-PCR step is depicted (mutated sites marked with red asterisks). Repeated icons are only labeled once. DNA fragments are represented as blue lines. T7, T7 RNA polymerase; BAC, bacterial artificial chromosome; YAC, yeast artificial chromosome; CPER, circular polymerase extension reaction; ISA, infectious subgenomic amplicons; CLEVER, CLoning-free and Exchangeable system for Virus Engineering and Rescue.

limitation of ISA (*Driouich et al., 2018*). The authors of this SuPReMe (Subgenomic Plasmids Recombination Method) version of ISA suggest omitting the PCR amplification step in favor of using subgenomic fragments directly liberated from sequence-verified plasmids. However, our optimized protocol states the exclusive use of high-fidelity polymerases and pooling of several parallel reactions, as well as a high template input and a reduced number of amplification cycles. To check on our optimized protocol, we performed NGS of eight reconstituted, recombinant viruses. The observed high fidelity – only one minor substitution was detected – was confirmed even in clonal virus populations: 3/5 viruses showed 100% sequence fidelity.

This low mutation rate was somehow not surprising as only high-fidelity polymerases were used, which are reported to have an error rate of $10^{-6}$ to $10^{-7}$ (*Xue et al., 2021*). Interestingly, very much underestimated in the field is the error rate of RNA polymerases used for in vitro transcription of

approximately $10^{-4}$ to $10^{-5}$ (**Brakmann and Grzeszik, 2001**), which consequently can lead to a highly polymorphic virus population. To our surprise, sequence integrity based on NGS data of recombinant virus was not always assessed (**Hou et al., 2020**), or only Sanger sequencing has been performed (**Torii et al., 2021**; **Xie et al., 2020**), while other studies show various unintentional mutations (**Amarilla et al., 2021**; **Rihn et al., 2021**).

A recent study on the versatile use of CPER introduces a confirmatory step by assembling the fragments in bacteria (circular polymerase extension cloning [CPEC]) and fully sequencing them prior to transfection (**Kim et al., 2023**). This introduced confirmatory step converts the straightforward and rapid protocol of the initially described CPER protocol to a tedious and time-consuming process. Further, analysis of reported mutations found within the 16 recombinant (mutant) viruses revealed several SNPs (**Kim et al., 2023**). Thus, the T7 polymerase transcription step might be responsible for the observed mutations within the viral populations.

We showed that successful recombination of a full-length cDNA copy and subsequent virus production was achieved in several individual cells, leading to a fitness-driven competition and therefore the eradication of any fitness-impairing mutations from the viral population. If clonal virus populations are desired, our established serial dilution protocol after transfection can be applied. Of note, we most probably did not yet reach the limits of the CLEVER platform during this study. We successfully tested fragment size from as big as 9 kb down to only 500 bp, and transfecting four or up to nine fragments (bigger/smaller or less/more fragments have not been tested). Combining these two observations, genomes much bigger than the investigated 30 kb could be reconstituted within the eukaryotic cell.

Finally, we demonstrate the full capacity of the CLEVER platform by direct mutagenesis of newly arising variants without any intermediate steps. Subgenomic fragments were directly amplified from viral RNA of Omicron BA.5 and XBB.1.5, whereas one primer pair was modified to delete ORF3a. The linker fragment harboring all heterologous sequences needed for intracellular DNA-dependent RNA transcription was added to the transfection, and recombinant Omicron BA.5 or XBB.1.5 virus with an ORF3a deletion, respectively, was rescued. There was no need of cloning or de novo synthesis of any BA.5 or XBB.1.5 sequences, allowing us to rapidly generate a mutant virus shortly after its emergence. Of note, this protocol focuses on rapid rescue and mutagenesis rather than high-sequence fidelity as key features of our optimized CLEVER protocol (such as the exclusive use of high-fidelity enzymes and pooling of several PCR reactions) were neglected in favor of a proof-of-concept attempt. Indeed, a slight increase in unintentional mutations was observed when sequencing clonal virus populations rescued from RNA directly (compare RNA clonal 1–3, **Supplementary file 2**).

As an additional feature, having the linker sequence in hand, extracted non-infectious viral RNA of emerging variants can be exchanged between laboratories with standard shipping conditions, and infectious virus can be subsequently rescued in BSL-3 conditions.

Along with the advantages of the CLEVER protocol, limitations must be considered: interestingly, virus was never rescued after transfecting Vero E6 cells, as already reported by others (**Mélade et al., 2022**). Whether this is due to low transfection efficiency or the cell's inability to recombine remains to be elucidated. Other cell lines not tested within this study will have to be tested for efficient recombination and virus production first. Further, the high sequence integrity of rescued virus is highly dependent on the fidelity of the DNA polymerase used for amplification. The use of other enzymes might negatively influence the sequence integrity of recombinant virus as it has been observed for the direct rescue from viral RNA using a commercially available one-step RT-PCR kit. Another limitation when performing direct mutagenesis is the synthesis of long oligos to create an overlapping region. Repetitive sequences, for example, can impair synthesis, and self-annealing and hairpin formation increase with prolonged oligos.

Taken together, CLEVER provides an elaborate platform for the rapid response to any newly emerging SARS-CoV-2 variant or the fast and efficient rescue of other plus-stranded RNA viruses. Virus can be directly rescued from any clinical isolate, and mutations such as SNPs or insertions/deletions of whole regions can be introduced without the need of any intermediate steps. Sequence integrity can be preserved when starting from sequence-verified plasmids rather than RNA directly. The CLEVER system thereby represents an excellent tool for studying large plus-stranded RNA viruses such as SARS-CoV-2, but also the studying of properties and gene functions in vitro of other, much smaller RNA viruses such as CHIKV or DENV, and for quickly adapting viral genomes to follow and study the evolution of any viral variants of clinical importance.

# Materials and methods

## Human samples

Human serum samples for neutralization assays were collected from SARS-CoV-2 vaccinated anonymous donors who gave their informed consent (approved by Ethikkommission Nordwest- und Zentralschweiz #2022–00303).

## Cells

BHK-21 and CHO-K1 cells were obtained from Thermo Scientific (ATCC CCL-10 and CCL-61). African green monkey kidney cells (Vero E6) were kindly provided by V. Thiel, Bern, Switzerland, and HEK29T cells were provided by D. Pinschewer, Basel, Switzerland. Adenocarcinomic human alveolar basal epithelial cells (A549) were obtained from NIBSC (A549-ACE-2 Clone 8-TMPRSS2; product number 101006) and HEK293 cells were obtained from Batavia (Cat# 110-025; Lot# 19D006).

Cells were cultivated in Dulbecco's modified Eagle medium (DMEM), high-glucose media (Cat# 1-26F50-I, BioConcept) supplemented with 10,000 U/mL of penicillin, 10 mg/mL of streptomycin (P/S) (Cat# 4-01F00-H, BioConcept), and 10% fetal bovine serum (FBS) (Cat# S0615, Sigma-Aldrich) at 37°C in a humidified atmosphere with 5% $CO_2$. Upon infection, cells were maintained in corresponding media with 2% FBS and cultured at 34°C. A lower incubation temperature was chosen based on previous studies (*V'kovski et al., 2021*).

Vero E6-TMPRSS2 cells were generated by transduction with a second-generation lentiviral vector pLEX307-TMPRSS2-blast (Addgene plasmid #158458) and selected for 2 wk in DMEM containing 20 µg/mL of Blasticidin (Cat# SBR00022, Sigma-Aldrich).

## Virus

Virus stocks of the initial Wuhan strain of SARS-CoV-2 were provided by G. Kochs, University of Freiburg, Germany (SARS-CoV_FR-3 [GenBank OR018857]; SARS-CoV_Muc [GenBank OR018856]).

SARS-CoV-2 Omicron variants BA.1 (GenBank OR018858), BA.5 (GenBank OR018859), and XBB.1.5 (GenBank OX393614) were isolated from nasopharyngeal aspirates of human anonymous donors, who had given their informed consent (approval by Ethikkommission Nordwest- und Zentralschweiz #2022-00303).

Clinical isolates of CHIKV, DENV1, and DENV3 were obtained from Dr. Karoline Leuzinger, University Hospital Basel, Switzerland.

All work including infectious SARS-CoV-2/CHIKV/DENV1/DENV3 viruses and their recombinant variants was conducted in a biosafety level 3 facility at the Department of Biomedicine within the University of Basel (approved by the Swiss Federal Office of Public Health (BAG) #A202850/3 and #A030187-2).

## Viral RNA extraction and cDNA conversion

Virus was propagated on Vero E6-TMPRSS2 cells and supernatant was harvested after 2 d of infection. CHIKV was propagated on BHK-21 cells and harvested after 1–2 d post-infection. DENV was propagated on Vero E6-TMPRSS2 cells and virus was harvested when a clear CPE was observed (5–7 d post-infection). RNA was extracted using the Maxwell RSC Viral Total Nucleic Acid Purification Kit (Cat# AS1330, Promega) or Maxwell RSC miRNA from Plasma or Serum (Cat# AS1680, Promega) following the manufacturer's protocol. Viral RNA was either used to prepare cDNA with the cDNA Synthesis Kit (Cat# BR0400401, biotechrabbit) or used directly as a template for RT-PCR using SuperScript IV One-Step RT-PCR System (Cat# 12594100, Invitrogen).

## DNA fragments for the generation of recombinant SARS-CoV-2

An adapted version of the ISA method (*Aubry et al., 2014*) was used to generate recombinant SARS-CoV-2. The genome was divided into four fragments (A–D) based on the reference sequence MT066156 (fragment A: nt1-nt8594; fragment B: nt8590-nt15107; fragment C: nt15100-nt20958; fragment D: nt20950-29867) and either amplified from a clinical isolate (FR-3 and Muc) and cloned into a modified pUC19 backbone (fragments A–D) or de novo synthesized (GenScript) and cloned into a pUC57 backbone (fragments A–C).

The human cytomegalovirus promoter (CMV) was cloned upstream of the DNA sequence of the viral 5′ UTR; herein, the first five nucleotides (ATATT) correspond to the 5′ UTR of SARS-CoV. Sequences corresponding to the poly(A) tail (n = 35), the hepatitis delta virus ribozyme (HDVr), and the simian virus 40 polyadenylation signal (SV40pA) were cloned immediately downstream of the DNA sequence of the viral 3′ UTR. Either eight or four fragments were PCR amplified with primers designed to generate ~100 bp overlap sequence between the adjacent fragments. Primers are listed in *Supplementary file 5*.

The Q5 High-Fidelity Polymerase kit (Cat# M0493, NEB) was used for amplification (5 µL 5× Q5 reaction buffer, 0.5 µM of each primer, 200 µM dNTP, 0.02 U/µl Q5 polymerase, 20–40 ng plasmid DNA, nuclease-free water up to 25 µL). PCR was performed on a Biometra thermocycler (TProfessional Trio) using the following conditions: initial denaturation at 98°C for 30 s, cycling at 98°C for 10 s, primer annealing (47–65°C) for 10 s, elongation at 72°C for 30 s/kb, followed by a final elongation at 72°C for 5 min. A two-step PCR program was applied: 5 cycles at low temperature according to the initial primer annealing site and 20 cycles at high annealing temperature (<65°C). Annealing temperatures and elongation times for the generation of recombinant SARS-CoV-2 using eight or four fragments are listed in *Supplementary file 6*.

To introduce mutations into the SARS-CoV-2 genome, primers were designed to create new homology regions for recombination while simultaneously introducing the mutation (summarized in *Figure 4—figure supplement 1*). Primer sequences to generate the fragments and introduce the desired mutations are listed in *Supplementary file 5*.

The size of PCR fragments was verified using gel electrophoresis and DNA was purified using the QIAquick PCR Purification Kit (Cat# 28104, QIAGEN). Concentration was measured using a Quantus Fluorometer (Promega) and the QuantiFluor ONE dsDNA System (Cat# E4871, Promega). When carryover of full-length plasmid cannot be excluded, a DpnI digestion step has to be considered. The fragments were mixed in an equimolar ratio, ethanol precipitated, and eluted to a final concentration of 1 µg/µL.

## Cloning-free reconstitution of SARS-CoV-2 directly from RNA using the linker fragment

Fragments were amplified directly from extracted viral RNA using the SuperScript IV One-Step RT-PCR System (Cat# 12594100, Invitrogen). Primers and PCR settings are listed in *Supplementary files 5 and 6*. An additional initial step at 50°C for 10 min was performed for cDNA conversion. For some virus reconstitutions, only one PCR reaction was performed per fragment and all fragments were pooled without further DNA quantification. In addition to the subgenomic fragments, a linker fragment comprising 100 bp overlap to fragment D2, the poly(A) tail (35), HDVr, SV40 followed by a spacer sequence, the CMV promoter and 100 bp of fragment A1 (total size of 1106 bp) was added. Then, 1 µg (if no DNA quantification of viral fragments was done) or a 5× molar excess of the linker fragment was added to the remaining fragments and transfected into HEK293T cells as described below.

### Linker fragment

The region comprising the 3′ termination signals (poly(A), HDVr, SV40 pA signal) and the last 100 bp of the 3′ UTR of SARS-CoV-2 as well as the region comprising the CMV promoter and the first 100 bp of the 5′ UTR of SARS-CoV-2 were amplified and inserted into pUC19 using Gibson assembly. The linker fragment was amplified using the primers listed in *Supplementary file 5*.

### Transfection and recovery of SARS-CoV-2

A total of 4–8 µg of an equimolar mix of purified DNA fragments (1 µg/µL) spanning the whole SARS-CoV-2 genome was transfected into $2 \times 10^6$ HEK293T cells using the SF Cell Line 4D-Nucleofector-X Kit (Cat# V4XC-2012, Lonza) and the 4D-Nucleofector X Unit (Lonza) with pulse code DS-150. The following pulse codes were used for different cell lines: CM-130 (A549-ACE2), DT-133 (CHO-K1), and CA-137 (BHK-21).

After pulsing, cuvettes were replenished with 1 mL RPMI-1640 (Cat# 1-41F51-I, BioConcept) and rested at 37°C, 5% $CO_2$ for 10 min before being transferred to 6-well plates filled with 3 mL of prewarmed DMEM 10% FBS, 1% P/S. Co-culture with Vero E6-TMPRSS2 cells was started 1–4 dpt and

the medium was changed to DMEM 2% FBS, 1% P/S. The supernatant of these co-cultures was transferred onto fresh Vero E6-TMPRSS2 cells and virus was harvested after CPE was observed (typically between 6 and 8 dpt).

For lipid-based transfection, HEK293T cells were seeded in 6-well plates in DMEM 10% FBS, 1% P/S, and grown overnight to reach 80–90% confluency. For JetPRIME (Cat# 101000001, Polyplus), 4 µg of DNA was transfected according to the manufacturer's protocol. For Lipofectamine 3000 (Cat# L3000015, Invitrogen), 6 µL Lipofectamine 3000 was diluted in 150 µL Opti-MEM (Cat# 31985070, Gibco) and mixed with 5 µg DNA, 10 µL P3000 Reagent diluted in 150 µL Opti-MEM, incubated for 15 min at room temperature (RT), and added to the cells. For Lipofectamine LTX & PLUS Reagent (Cat# 15338100, Invitrogen), 12 µL Lipofectamine LTX was diluted in 150 µL Opti-MEM and mixed with 4 µg DNA, 4 µL PLUS reagent diluted in 150 µL Opti-MEM, incubated for 5 min at RT, and added to the cells.

Successful virus rescue was monitored with SARS-CoV-2 Rapid Antigen Test (Cat# 9901-NCOV-01G, SD BIOSENSOR, Roche) by applying 100 µL of unfiltered culture supernatant directly onto the test device.

## Cloning-free rescue of CHIKV and DENV1/DENV3

Overlapping fragments spanning the whole CHIKV (three fragments) or DENV1/DENV3 (two fragments) genome were amplified directly from extracted viral RNA as described above. The linker fragment was amplified with specific primers for CHIKV, DENV1, or DENV3, respectively, to introduce a homology region to the viral genome. Primers and PCR settings are listed in *Supplementary files 5 and 6*. To prevent recovery of wild-type virus from transfected total RNA, PCR fragments were digested with RNase A (Cat# EN0531, Thermo Scientific) before transfection. Additionally, a marker SNP was introduced within the recombination sites to differentiate between recombinant and wild-type viruses. A total of 6 µg DNA, including a 5× molar excess of the corresponding linker fragment, was transfected into BHK-21 cells using electroporation (see above). For CHIKV, CPE was observed as early as 2 dpt on BHK-21 cells. For DENV1 and DENV3, supernatant from transfected BHK-21 cells was transferred onto Vero E6-TMPRSS2 cells 6 dpt to observe CPE formation. Virus was passaged twice and the introduced marker SNP was confirmed by Sanger sequencing (*Supplementary file 4*).

## Growth kinetics

Vero E6 cells were seeded at 50% confluency in 12-well plates and cultured at 37°C, 5% $CO_2$ in DMEM supplemented with 10% FBS, 1% P/S for 20 hr. The medium was replaced by DMEM 2% FBS, 1% P/S prior to inoculation with an MOI of 0.01 of reconstituted SARS-CoV-2 or clinical isolate in a total volume of 0.5 mL. Cell layers were washed with PBS 2 hpi and 2 mL of medium was added. Infections were set up in triplicates. Also, 200 µL were immediately collected for measuring input material (time point 0). Then, 12, 24, 48, and 72 hpi, 200 µL of supernatant were collected to determine viral titers. The time-course experiment was done in three biological replicates starting with the same input material. All samples were stored at –80°C until analysis. Infectious titers were determined by plaque assay.

## Next-generation sequencing

NGS was performed at Seq-IT GmbH & Co. KG Kaiserslautern using EasySeq SARS-CoV-2 WGS Library Prep Kit (NimaGen, SKU: RC-COV096) according to the manufacturer's protocol. Sequencing was performed using a 300-cycle Mid Output kit on a NextSeq 500 system.

## Standard plaque assay

Viral titers were determined by counting plaque-forming units (PFU) after incubation on susceptible cells. Vero E6 cells were seeded at a density of $4 \times 10^6$ cells/96-well flat-bottom plate in DMEM/2% FBS and incubated overnight at 37°C and 5% $CO_2$. Virus was added 1:10 onto the cell monolayer in duplicates or triplicates and serially diluted 1:2 or 1:3. Plates were incubated for 2 d at 34°C, 5% $CO_2$ until plaque formation was visible. For virus inactivation, 80 µL of formaldehyde (15% w/v in PBS; Cat# F8775, Sigma-Aldrich) was added for 10 min to the cultures. After this period, fixative and culture medium were aspirated, and crystal violet (0.1% w/v) was added to each well and incubated for 5 min. Afterward, the fixed and stained plates were gently rinsed several times under tap water and dried prior to analysis on a CTL ImmunoSpot analyzer.

## Immunoblotting

Vero E6 cells were seeded at confluency in 6-well plates and infected with an MOI of 0.5. Then, 30 hpi, after the first sights of CPE were visible, cells were washed and lysed in 1× Laemmli buffer. Lysates were boiled at 95°C for 5 min before being loaded onto a polyacrylamide gel (Cat# 4561094, Bio-Rad). Proteins were blotted onto a PVDF membrane (Bio-Rad, Cat# 1704156) using Trans-Blot Turbo Transfer (Bio-Rad) and blocked with 1% BSA in PBS with 0.05% Tween-20 for 1 hr. Primary antibody was diluted in PBS, 0.05% Tween-20, 1% BSA, and incubated on a shaker at 4°C for 16 hr. The membrane was washed three times in PBS, 0.05% Tween-20 before the secondary antibody was diluted in PBS, 0.05% Tween-20, 1% BSA, and incubated on a shaker for 1 hr at RT in the dark. The membrane was washed four times for 10 min in PBS, 0.05% Tween-20. Signals were acquired using an image analyzer (Odyssey CLx imaging system, LI-COR). Antibodies are listed below.

## Immunocytochemistry

For co-localization studies and validation of knock-out viral constructs, ~1 × 10$^5$ Vero E6 cells were seeded onto glass coverslips in 24-well plates and grown overnight. Cells were infected in low medium volume for 2 hr before the medium was added to the final volume of 1 mL and incubated for 24 hr. Cells were fixed with 4% PFA in PBS for 10 min at RT, washed, and subsequently stained. Briefly, cells were blocked with 10% Normal Donkey Serum (Cat# 017-000-121, Jackson ImmunoResearch) and 0.1% Triton X-100 for 30 min followed by incubation with primary antibodies for 1 hr in 1% Normal Donkey Serum, 1% BSA, and 0.3% Triton X-100 in PBS. Cells were washed three times 10 min with 1× PBS, 0.1% BSA, and incubated with secondary antibodies for 1 hr in 1% Normal Donkey Serum, 1% BSA, and 0.3% Triton X-100 in PBS. Cells were washed once with 1× PBS, 0.1% BSA, and washed three times with 1× PBS before mounting on microscope slides using Fluoromount-G (Cat# 0100-01, SouthernBiotech). Hoechst 33342 dye (Cat# B2261, Sigma-Aldrich) was co-applied during washing at a final concentration of 0.5 µg/mL for nuclear staining.

Images from co-localization studies and knock-out validation were acquired on an inverted spinning-disk confocal microscope (Nikon Ti2 equipped with a Photometrics Kinetix 25 mm back-illuminated sCMOS, Nikon NIS AR software), using ×40 Plan-Apochromat objectives (numerical aperture 0.95) and were then processed in Fiji and Omero.

## Antibodies

The following antibodies were used in this study: mouse monoclonal anti-β-actin (Cell Signaling Technology; Cat# 3700; RRID:AB_2242334; Lot# 20), rabbit polyclonal anti-FLAG (Cell Signaling Technology; Cat# 14793; RRID:AB_2572291; Lot# 5), rat monoclonal anti-FLAG (BioLegend; Cat# 637301; RRID:AB_1134266; Lot# B318853), rabbit polyclonal anti-SARS-CoV-2 NSP2 (GeneTex; Cat# GTX135717; RRID:AB_2909866; Lot# B318853), mouse monoclonal anti-SARS-CoV-2 Nucleocapsid protein (4F3C4, gift from Sven Reiche; *Bussmann et al., 2006*), sheep polyclonal anti-SARS-CoV-2 ORF3a (*Rihn et al., 2021*), and rabbit polyclonal anti-SARS-CoV-2 ORF8 (Novus Biologicals; Cat# NBP3-07972; Lot# 25966-2102).

Fluorophore-conjugated secondary antibodies were from Jackson ImmunoResearch (Cy3 donkey anti-rat #712-165-153, Cy3 donkey anti-mouse #715-165-151, Cy5 donkey anti-rabbit #711-175-152, Cy5 donkey anti-sheep #713-175-147), Li-Cor (IRDye 680RD donkey anti-mouse #926-68072, IRDye 680RD goat anti-rabbit #926-68071), and Invitrogen (Alexa Fluor 680 donkey anti-sheep #A21102).

## Cryo-transmission electron microscopy (cryo-TEM)

Virus particles were fixed in 1% glutaraldehyde (Cat# 233281000, Thermo Scientific). A 4 µL aliquot of sample was adsorbed onto holey carbon-coated grid (Lacey, Tedpella, USA), blotted with Whatman 1 filter paper, and vitrified into liquid ethane at –180°C using a Leica GP2 plunger (Leica Microsystems, Austria). Frozen grids were transferred onto a Talos 200C Electron microscope (FEI, USA) using a Gatan 626 cryo-holder (Gatan, USA). Electron micrographs were recorded at an accelerating voltage of 200 kV using a low-dose system (40 e-/Å2) and keeping the sample at –175°C. Defocus values were –2 to 3 µm. Micrographs were recorded on 4K × 4K Ceta CMOS camera.

## Sanger sequencing of recombinant virus

RNA was extracted as described above and the region of interest was amplified using specific primer pairs and the SuperScript IV One-Step RT-PCR System (Cat# 12594100, Invitrogen). Amplified region was directly sent for overnight sequencing service at Microsynth AG, Balgach, Switzerland.

## Neutralization assay

Vero E6 cells were seeded in 96-well flat-bottom plates, $3.5 \times 10^6$ cells/plate in a final volume of 100 µL DMEM complemented with 2% FBS, 1% P/S. Cells were incubated at 37°C, 5% $CO_2$ overnight to reach confluency. Patient sera were serially diluted 1:2 in a 96-well round-bottom plate, starting with a 1:20 dilution. Virus was added to the diluted sera at a final MOI of 0.002 per well and incubated for 1 hr at 34°C, 5% $CO_2$. Pre-incubated sera/virus was added to the cells and incubated for 3 d at 34°C, 5% $CO_2$. Cells were fixed as described above (Standard Plaque Assay).

## Quantification and statistical analysis

Statistical analysis was conducted with GraphPad Prism 9. Sample sizes were chosen based on previous experiments and literature surveys. No statistical methods were used to predetermine sample sizes. Appropriate statistical tests were chosen based on sample size and are indicated in individual experiments.

## Materials and correspondence

Further information and requests for resources and reagents should be directed to and will be fulfilled by the lead contact, Thomas Klimkait (thomas.klimkait@unibas.ch).

## Acknowledgements

We thank Dr. G Kochs, Freiburg, D, for providing the SARS-CoV-2 Wuhan isolate, Piet for providing the SARS-CoV-2 isolate Omicron BA.1, and the laboratory of Dr. A Egli (University Hospital Basel, CH) and M Daeumer and A Thielen (SeqIT GmbH, Kaiserslautern, D) for NGS sequencing service. We thank Dr. Karoline Leuzinger, University Hospital Basel, CH, for providing clinical isolates of CHIKV and DENV1/3. Figures were created with BioRender.com. This research was co-funded through a federal project grant via Innosuisse project #52533.1 IP-LS and by RocketVax through a project grant with USB/Canton BS. The funders had no influence on the experimental design or the conduct of work.

## Additional information

### Funding

| Funder | Grant reference number | Author |
|---|---|---|
| Innosuisse - Schweizerische Agentur für Innovationsförderung | #52533.1 IP-LS | Thomas Klimkait |
| University Hospital Basel / Canton BS and RocketVax | | Thomas Klimkait |

The funders had no role in study design, data collection and interpretation, or the decision to submit the work for publication.

### Author contributions

Enja Tatjana Kipfer, Conceptualization, Data curation, Formal analysis, Validation, Investigation, Visualization, Methodology, Writing – original draft, Project administration, Writing – review and editing; David Hauser, Conceptualization, Data curation, Formal analysis, Investigation, Visualization, Writing – review and editing; Martin J Lett, Fabian Otte, Lorena Urda, Yuepeng Zhang, Investigation, Writing – review and editing; Christopher MR Lang, Mohamed Chami, Investigation; Christian Mittelholzer, Writing – original draft, Writing – review and editing; Thomas Klimkait, Conceptualization, Supervision, Funding acquisition, Project administration, Writing – review and editing

### Author ORCIDs
Enja Tatjana Kipfer https://orcid.org/0000-0001-5870-6784
David Hauser http://orcid.org/0000-0001-9950-365X
Martin J Lett https://orcid.org/0000-0002-7230-9264
Fabian Otte http://orcid.org/0000-0003-0572-617X
Lorena Urda http://orcid.org/0009-0007-4122-515X
Yuepeng Zhang http://orcid.org/0009-0009-7923-2173
Mohamed Chami http://orcid.org/0000-0002-8733-055X
Christian Mittelholzer http://orcid.org/0000-0003-4353-3555
Thomas Klimkait http://orcid.org/0000-0003-4945-6511

### Ethics
Human serum samples for neutralization assays were collected from SARS-CoV-2 vaccinated anonymous donors who gave their informed consent (approved by Ethikkommission Nordwest- und Zentralschweiz #2022-00303).

Joint Public Review: https://doi.org/10.7554/eLife.89035.3.sa1
Author Response https://doi.org/10.7554/eLife.89035.3.sa2

## Additional files

### Supplementary files
• Supplementary file 1. Infectivity assessment of recombinant virus on Vero E6 cells. Vero E6 cells were infected with the indicated recombinant virus and pictures were taken 3 d post-infection. Of note, the virus was not titrated and the development of CPE is not quantitative. Scale bar represents 100 μm.

• Supplementary file 2. Genomic characterization of recombinant SARS-CoV-2 virus based on NGS data. Mutations with a relative abundance of >10% in the entire virus population are listed. Ambiguities or low coverage are highlighted in italics.

• Supplementary file 3. Homology regions successfully used for recombinant SARS-CoV-2, CHIKV, and DENV1/DENV3. Length, GC content, and the hypothetical annealing temperature (according to OligoCalc, salt-adjusted) are listed. Note that homology regions were chosen independently from GC content or annealing temperature and values are only listed for completion.

• Supplementary file 4. Sanger sequencing data of the region of interest for the generated mutant rSARS-CoV-2, rCHIKV, and rDENV1/rDENV3.

• Supplementary file 5. Oligonucleotide list.

• Supplementary file 6. PCR settings for individual fragments.

• Supplementary file 7. Plasmids.

• MDAR checklist

### Data availability
Raw sequencing data are deposited in the Sequence Read Archive of the National Center for Biotechnology Information (NCBI) with accession number PRJNA971789. All other data generated or analysed during this study are included in the manuscript and supporting files. Source data files have been provided for Figures 1, 4, and 5 and Figure 5—figure supplements 1 and 2.

The following dataset was generated:

| Author(s) | Year | Dataset title | Dataset URL | Database and Identifier |
|---|---|---|---|---|
| Kipfer ET, Hauser D, Lett MJ, Otte F, Urda L, Zhang Y, Lang CMR, Chami M, Mittelholzer C, Klimkait T | 2023 | Next generation sequencing of recombinant SARS-CoV-2 virus | https://www.ncbi.nlm. nih.gov/sra/?term= PRJNA971789 | NCBI Sequence Read Archive, PRJNA971789 |

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

# Appendix 1

**Appendix 1—key resources table**

| Reagent type (species) or resource | Designation | Source or reference | Identifiers | Additional information |
|---|---|---|---|---|
| Strain, strain background (*Escherichia coli*) | DH5α *E. coli* | NEB | Cat# C2987 | Competent cells |
| Strain, strain background (*E. coli*) | One Shot Stbl3 Chemically Competent *E. coli* | Invitrogen | Cat# C737303 | Competent cells |
| Strain, strain background (*SARS-CoV-2*) | SARS-CoV-2 Wuhan (SARS-CoV_FR-3) | Other | GenBank OR018857 | Provided by G. Kochs, Freiburg, Germany |
| Strain, strain background (*SARS-CoV-2*) | SARS-CoV-2 Wuhan (SARS-CoV_Muc) | Other | GenBank OR018856 | Provided by G. Kochs, Freiburg, Germany |
| Strain, strain background (*SARS-CoV-2*) | SARS-CoV-2 Omicron variant BA.1 | Human donors | GenBank OR018858 | Approved by Ethikkommission Nordwest- und Zentralschweiz #2022-00303 |
| Strain, strain background (*SARS-CoV-2*) | SARS-CoV-2 Omicron variant BA.5 | Human donors | GenBank OR018859 | Approved by Ethikkommission Nordwest- und Zentralschweiz #2022-00303 |
| Strain, strain background (*SARS-CoV-2*) | SARS-CoV-2 Omicron variant XBB.1.5 | Human donors | GenBank OX393614 | Approved by Ethikkommission Nordwest- und Zentralschweiz #2022-00303 |
| Strain, strain background (*CHIKV*) | Chikungunya virus (CHIKV) | Other | – | Provided by K. Leuzinger, Basel, Switzerland |
| Strain, strain background (*DENV1*) | Dengue virus serotype 1 (DENV1) | Other | – | Provided by K. Leuzinger, Basel, Switzerland |
| Strain, strain background (*DENV3*) | Dengue virus serotype 3 (DENV3) | Other | – | Provided by K. Leuzinger, Basel, Switzerland |
| Cell line (*Chlorocebus sabaeus*) | African green monkey kidney cells (Vero E6) | Other | – | Provided by V. Thiel, Bern, Switzerland |
| Cell line (*Homo sapiens*) | Adenocarcinomic human alveolar basal epithelial cells (A549) | NIBSC | Cat# 101006 | – |
| Cell line (*Mesocricetus auratus*) | Baby hamster kidney cells (BHK-21) | Thermo Scientific | ATCC CCL-10 | – |
| Cell line (*H. sapiens*) | Human embryonic kidney cells (HEK293T) | Other | – | Provided by D. Pinschewer, Basel, Switzerland |
| Cell line (*H. sapiens*) | Human embryonic kidney cells (HEK293) | Batavia | Cat# 110-025; Lot# 19D006 | – |
| Cell line (*Cricetulus griseus*) | Chinese hamster ovary cells (CHO-K1) | Thermo Scientific | ATCC CCL-61 | – |

*Appendix 1 Continued on next page*

*Appendix 1 Continued*

| Reagent type (species) or resource | Designation | Source or reference | Identifiers | Additional information |
|---|---|---|---|---|
| Biological sample (*H. sapiens*) | Serum samples | Human donors | – | Approved by Ethikkommission Nordwest- und Zentralschweiz #2022-00303 |
| Antibody | Anti-β-actin (mouse monoclonal) | Cell Signaling Technology | Cat# 3700; RRID:AB_2242334; Lot# 20 | WB (1:1000) |
| Antibody | Anti-FLAG (rabbit polyclonal) | Cell Signaling Technology | Cat# 14793; RRID:AB_2572291; Lot# 5 | WB (1:1000) |
| Antibody | Anti-FLAG (rat monoclonal) | BioLegend | Cat# 637301; RRID:AB_1134266; Lot# B318853 | ICC (1:1000) |
| Antibody | Anti-SARS-CoV-2 NSP2 (rabbit polyclonal) | GeneTex | Cat# GTX135717; RRID:AB_2909866; Lot# B318853 | WB (1:5000) |
| Antibody | Anti-SARS-CoV-2 Nucleocapsid protein (mouse monoclonal, 4F3C4) | Sven Reiche (doi: 10.1016/j.virusres.2006.07.005) | – | ICC (1:500) |
| Antibody | Anti-SARS-CoV-2 ORF3a (sheep polyclonal) | MRC PPU reagents (doi:10.1371/journal.pbio.3001091) | – | WB (1:1000) ICC (1:500) |
| Antibody | Anti-SARS-CoV-2 ORF8 (rabbit polyclonal) | Novus Biologicals | Cat# NBP3-07972; Lot# 25966-2102 | WB (1:1000) ICC (1:1000) |
| Recombinant DNA reagent | pLEX307-TMPRSS2-blast (plasmid) | Addgene | Cat# 158458 | – |
| Recombinant DNA reagent | pUC19_CoV-2_Linker | This study | Addgene plasmid #211731 | Plasmid expressing 5' and 3' regions for direct rescue |
| Recombinant DNA reagent | pUC19_CoV-2_frA (plasmid) | This study | – | Plasmid encoding SARS-CoV-2 'fragment A,' see ***Supplementary file 7*** |
| Recombinant DNA reagent | pUC19_CoV-2_frB (plasmid) | This study | – | Plasmid encoding SARS-CoV-2 'fragment B,' see ***Supplementary file 7*** |
| Recombinant DNA reagent | pUC19_CoV-2_frC (plasmid) | This study | – | Plasmid encoding SARS-CoV-2 'fragment C,' see ***Supplementary file 7*** |
| Recombinant DNA reagent | pUC19_CoV-2_frD (plasmid) | This study | – | Plasmid encoding SARS-CoV-2 'fragment D' Wuhan isolate, see ***Supplementary file 7*** |
| Recombinant DNA reagent | pUC19_CoV-2_frD_S Omicron BA.1 (plasmid) | This study | – | Plasmid encoding SARS-CoV-2 'fragment D' Omicron BA.1 isolate, see ***Supplementary file 7*** |
| Recombinant DNA reagent | pUC19_CoV-2_frD_S Omicron BA.5 (plasmid) | This study | – | Plasmid encoding SARS-CoV-2 'fragment D' Omicron BA.5 isolate, see ***Supplementary file 7*** |
| Recombinant DNA reagent | pUC57_CoV-2_frA (plasmid) | GenScript | – | High-quality DNA ordered from GenScript |
| Recombinant DNA reagent | pUC57_CoV-2_frB (plasmid) | GenScript | – | High-quality DNA ordered from GenScript |

*Appendix 1 Continued on next page*

*Appendix 1 Continued*

| Reagent type (species) or resource | Designation | Source or reference | Identifiers | Additional information |
|---|---|---|---|---|
| Recombinant DNA reagent | pUC57_CoV-2_frC (plasmid) | GenScript | – | High-quality DNA ordered from GenScript |
| Recombinant DNA reagent | SARS-CoV-2 Omicron Strain S gene original_pcDNA3.1(+) | GenScript | Cat# MC_0101273 | – |
| Sequence-based reagent | Oligonucleotides (primers) | This study | – | For all primers, see **Supplementary file 5** |
| Commercial assay or kit | Maxwell RSC Viral Total Nucleic Acid Purification Kit | Promega | Cat# AS1330 | – |
| Commercial assay or kit | Maxwell RSC miRNA from Plasma or Serum | Promega | Cat# AS1680 | – |
| Commercial assay or kit | SuperScript IV One-Step RT-PCR System | Invitrogen | Cat# 12594100 | – |
| Commercial assay or kit | SF Cell Line 4D-Nucleofector-X Kit | Lonza | Cat# V4XC-2012 | – |
| Commercial assay or kit | EasySeq SARS-CoV-2 WGS Library Prep Kit | NimaGen | SKU: RC-COV096 | – |
| Chemical compound, drug | Blasticidin | Sigma-Aldrich | Cat# SBR00022 | – |
| Chemical compound, drug | Hoechst 33342 dye | Sigma-Aldrich | Cat# B2261 | – |
| Chemical compound, drug | RNase A | Thermo Scientific | Cat# EN0531 | – |
| Software, algorithm | ImageJ v2.9.0/1.53t | NHI | https://imagej.nih.gov/ij/download.html; RRID:SCR_003070 | – |
| Software, algorithm | Prism v9 | GraphPad | https://www.graphpad.com/scientific-software/prism/; RRID:SCR_002798 | – |
| Software, algorithm | Omero | Open Microscopy Environment | http://www.openmicroscopy.org/site/products/omero; RRID:SCR_002629 | – |
| Software, algorithm | Adobe Illustrator CC | Adobe | http://www.adobe.com/products/illustrator.html; RRID:SCR_010279 | – |
| Software, algorithm | 7500 Real-Time PCR Software Version 2.0.6 | Applied Biosystems | RRID:SCR_014596 | – |
| Software, algorithm | ImmunoSpot Software Version 7.0.26.0 | ImmunoSpot | RRID:SCR_011082 | – |
| Software, algorithm | Nikon NIS-Elements AR Versions 5.30.07 | Nikon | RRID:SCR_014329 | – |
| Software, algorithm | LI-COR Image Studio Version 2.0 | LI-COR | – | – |
| Other | SARS-CoV-2 Rapid Antigen Test | Roche | Cat# 9901-NCOV-01G | SARS-CoV-2 Antigen Test to check for viral rescue, see **Figure 1** |

