## [Editor Report · eLife assessment]

This study describes CLEVER, an improved method for fast and efficient rescue and mutagenesis of SARS-CoV-2. While the principle of this method is not new, this work significantly improves upon existing protocols, providing an **important** advancement in the field of viral infectious clones. **Convincing** proof-of-concept experiments were performed that demonstrate the utility and efficiency of the method.

---

## [Referee Report · Joint Public Review]

In this manuscript, Kipfer et al describe a method for a fast and accurate SARS-CoV-2 rescue and mutagenesis. This work is based on a published method termed ISA (infectious subgenomic amplicons), in which partially overlapping DNA fragments covering the entire viral genome and additional 5' and 3' sequences are transfected into mammalian cell lines. These DNA fragments recombine in the cells, express the full length viral genomic RNA and launch replication and rescue of infectious virus.

CLEVER, the method described here significantly improves on the ISA method to generate infectious SARS-CoV-2, making it widely useful to the virology community.

Specifically, the strengths of this method are:

1. The successful use of various cell lines and transfection methods.

2. Generation of a four-fragment system, which significantly improves the method efficiency due to lower number of required recombination events.

3. Flexibility in choice of overlapping sequences, making this system more versatile.

4. The authors demonstrated how this system can be used to introduce point mutations as well as insertion of a tag and deletion of a viral gene.

5. Fast-tracking generation of infectious virus directly from RNA of clinical isolates by RT-PCR, without the need for cloning the fragments or using synthetic sequences.

6. The authors further expanded this method to work on additional plus-strand RNA viruses beyond SARS-CoV-2 (CHIKV, DENV)

The manuscript clearly presents the findings, and the proof-of-concept experiments are well designed.

Overall, this is a very useful method for SARS-CoV2 research. Importantly, it can be applicable to many other viruses, speeding up the response to newly emerging viruses than threaten the public health.

---

## [Author Response]

The following is the authors’ response to the original reviews.

We are very grateful to the reviewers for their thoughtful comments on the manuscript and to the editors for their assessment.

We thank the reviewers for their positive feedback and appreciate that they consider our method a valid addition to previously established systems for generating recombinant RNA viruses.

To strengthen this point, we have now included additional validation by the rescue of recombinant Chikungunya and Dengue virus from viral RNA directly, using the CLEVER protocol. This strengthens the potential of this method as a reverse genetics platform for positive-stranded viruses in general.

The supportive data has been amended in the Results section, taken into account in Materials and Methods, and the corresponding supplementary figure (Figure S4) has been added.

One key point raised by one of the reviewers, a comparison with different systems, could not be addressed in this manuscript as our lab does not at all perform BAC cloning. We currently do not have the necessary expertise to conduct an unbiased side-by-side comparison.

All other comments were addressed in detail, either by including additional data or through specific clarification in the revised text. We are grateful for the careful review and constructive criticisms raised by the reviewers and feel that the corrections and additions have significantly improved the manuscript.

We have revised the latest version posted May 30, 2023 on bioRxiv (https://doi.org/10.1101/2023.05.11.540343).

**Reviewer #1:**

**Public Review:**
In this manuscript, Kipfer et al describe a method for a fast and accurate SARS-CoV2 rescue and mutagenesis. This work is based on a published method termed ISA (infectious subgenomic amplicons), in which partially overlapping DNA fragments covering the entire viral genome and additional 5' and 3' sequences are transfected into mammalian cell lines. These DNA fragments recombine in the cells, express the full length viral genomic RNA and launch replication and rescue of infectious virus.CLEVER, the method described here significantly improves on the ISA method to generate infectious SARS-CoV2, making it widely useful to the virology community.Specifically, the strengths of this method are:1. The successful use of various cell lines and transfection methods.1. Generation of a four-fragment system, which significantly improves the method efficiency due to lower number of required recombination events.1. Flexibility in choice of overlapping sequences, making this system more versatile.1. The authors demonstrated how this system can be used to introduce point mutations as well as insertion of a tag and deletion of a viral gene.1. Fast-tracking generation of infectious virus directly from RNA of clinical isolates by RT-PCR, without the need for cloning the fragments or using synthetic sequences.One weakness of the latter point, which is also pointed out by the authors, is that the direct rescue of clinical isolates was not tested for sequence fidelity.The manuscript clearly presents the findings, and the proof-of-concept experiments are well designed.Overall, this is a very useful method for SARS-CoV2 research. Importantly, it can be applicable to many other viruses, speeding up the response to newly emerging viruses than threaten the public health.

We thank the reviewer for this positive feedback and the summary of the main points. Nevertheless, we would like to comment on point (5): “the direct rescue of clinical isolates was not tested for sequence fidelity”

This impression by the reviewer suggests that the data was not sufficient on this point. However, the sequence fidelity after direct rescue from RNA was indeed tested in this study, even on a clonal level (please see: Table S2, or raw NGS data SRX20303605 - SRX20303607).For higher clarity, we added the following sentence to the manuscript:

“Indeed, a slight increase of unintentional mutations was observed when sequencing clonal virus populations rescued from RNA directly”.

**Recommendations for the authors:**
Minor Points:1. On page 8, the authors write: "levels correlated very well with the viral phenotype". This sentence is not clear. Please clarify what you mean by "viral phenotype". Do you mean CPE on Vero cells?

We corrected the sentence to: “(…) staining intensity and patterns correlated very well with the wild-type phenotype.”

1. Page 9 "sequences were analyzed with a cut-off of 10%. Cutoff of what? please clarify.

The sentence was rephrased to: “(…)mutations with a relative abundance of >10% in the entire virus population were analyzed”

1. Page 15: The authors refer to the time required for completion of each step of the process. It would be helpful and informative for the readers to include a panel in figure 4, visualizing the timelines.

We included a timeline in Figure 4, Panel A.

1. Materials and methods, first paragraph: Please specify which human samples were collected. Do the authors refer to clinical virus isolates?

We added the following information to the Materials and Methods section:

“Human serum samples for neutralization assays were collected from SARS-CoV-2 vaccinated anonymous donors (…)”

Clinical virus isolates (Material and Methods; Virus) were used for control experiments, neutralization assays, or as templates for RT-PCR.

1. Supplementary figure 4A: The color scheme makes it hard to differentiate between the BA.1 and BA.5 fragments. Please choose colors that are not as similar to each other.

Colors were adapted for better distinction.

**Reviewer #2:**
Public Review:The authors of the manuscript have developed and used cloning-free method. It is not entirely novel (rather it is based on previously described ISA method) but it is clearly efficient and useful complementation to the already existing methods. One of strong points of the approach use by authors is that it is very versatile, i.e. can be used in combination with already existing methods and tools. I find it important as many laboratories have already established their favorite methods to manipulate SARS-CoV-2 genome and are probably unwilling to change their approach entirely. Though authors highlight the benefits of their method these are probably not absolute - other methods may be as efficient or as fast. Still, I find myself thinking that for certain purposes I would like to complement my current approach with elements from authors CLEVER method.The work does not contain much novel biological data - which is expected for a paper dedicated to development of new method (or for improving the existing one). It may be kind of shortcoming as it is commonly expected that authors who have developed new methods apply it for discovery of something novel. The work stops on step of rescue the viruses and confirming their biological properties. This part is done very well and represents a strength of the study. The properties of rescued viruses were also studied using NSG methods that revealed high accuracy of the used method, which is very important as the method relies on use of PCR that is known to generate random mistakes and therefore not always method of choice.What I found missing is a real head-to-head comparison of the developed system with an existing alternatives, preferably some PCR-free standard methods such as use of BAC clones. There are a lot of comparisons but they are not direct, just data from different studies has been compared. Authors could also be more opened to discuss limitations of the method. One of these seems to be rather low rescue efficiency - 1 rescue event per 11,000 transfected cells. This is much lower compared to infectious plasmid (about 1 event per 100 cells or so) and infectious RNAs (often 1 event per 10 cells, for smaller genomes most of transfected cells become infected). This makes the CLEVER method poorly suitable for generation of large infectious virus libraries and excludes its usage for studies of mutant viruses that harbor strongly attenuating mutations. Many of such mutations may reduce virus genome infectivity by 3-4 orders of magnitude; with current efficiencies the use of CLEVER approach may result in false conclusions (mutant viruses will be classified as non-viable while in reality they are just strongly attenuated).

We thank reviewer 2 for the careful review of our work and the valuable feedback. We agree that a direct comparison with other (PCR-free) methods such as BAC cloning, could be useful for demonstrating the unique benefits of the CLEVER method. However, as our laboratory does not use any BAC or YAC cloning methods, we could not ensure an unbiased side-byside comparison using different techniques.

We would like to highlight the avoidance of any yeast/bacterial cloning steps that render the CLEVER protocol significantly faster and easier to handle. A visualization of the key steps that could be skipped using CLEVER in comparison to common reverse genetics methods is given in Figure 6.

Further, we firmly believe that the benefits of the CLEVER method become especially apparent for large viral genomes such as the one of SARS-CoV-2, where assembly, genome amplification and sequence verification of plasmid DNA are highly inefficient and more timeconsuming than for small viruses like DENV, CHIKV or HIV.

We agree with the reviewer that the overall transfection and recombination efficiencies observed with CLEVER seemed rather low. Although data on transfection/rescue efficiency is known for many techniques and viruses, we did not find any published data on the reconstitution of SARS-CoV-2 or viruses with similar genome sizes. Therefore, a useful comparator for our observations in relation to other techniques is currently simply missing. We therefore emphasize that the efficiencies of CLEVER were achieved with one of the largest plus-stranded RNA virus genomes, and our data can’t be directly compared to transfection efficiencies of short infectious RNAs.

On the contrary, it was rather interesting to observe the very high rescue efficiency of infectious virus progeny. During the two years of establishing and validating the CLEVER protocol, we reached success rates for the genome reconstitution after transfection of >95 %. This was even obtained with highly attenuated mutants including rCoV2∆ORF3678 (joint deletion of ORF3a, ORF6, ORF7a, and ORF8) (Liu et al., 2022)(see Author response image 1). We amended this data in response to the reviewers’ comment and as an example of the successful rescue of an attenuated virus from five overlapping genome fragments (fragments A, B, C, D1, and D2∆ORF3678).

The latter data were not added to the main manuscript since in this case the deletions were introduced using a different method: from the plasmid-based DNA fragment D2∆ORF3678 and not directly from PCR-based mutagenesis.

Further, CLEVER was used for related substantial manipulations, including the complete deletion of the Envelope gene (E) which led to the creation of a single-cycle virus that may serve as a live, replication-incompetent vaccine candidate (Lett et al., 2023).

**Author response image 1. sa2fig1:** rCoV2∆ORF3678. Detection of intracellular SARS-CoV-2 nucleocapsid protein (N, green) and nuclei (Hoechst, blue) in Vero E6TMPRSS2 cells infected with rCoV2∆ORF3678 by immunocytochemistry. Scalebar is 200 µm in overview and 50 µm in ROI images.

Recommendations for the authors:The work is nicely presented and the method authors has developed is clearly valuable. As indicated in Public review section the work would benefit from direct comparison of CLEVER with that of infectious plasmid (or RNA) based methods; direct comparison of data would be more convincing that indirect one. Authors should also discuss possible limitations of the method - this is helpful for a reader.

We were not able to perform a direct comparison of CLEVER with other methods (see our statement above).

We added the following section to the discussion: “Along with the advantages of the CLEVER protocol, limitations must be considered: Interestingly, virus was never rescued after transfecting Vero E6 cells, as has been observed previously (Mélade et al., 2022). Whether this is due to low transfection efficiency or the cell’s inability to recombine remains to be elucidated. Other cell lines not tested within this study will have to be tested for efficient recombination and virus production first. Further, the high sequence integrity of rescued virus is highly dependent on the fidelity of the DNA polymerase used for amplification. The use of other enzymes might negatively influence the sequence integrity of recombinant virus, as it has been observed for the direct rescue from viral RNA using a commercially available onestep RT-PCR kit. Another limitation when performing direct mutagenesis is the synthesis of long oligos to create an overlapping region. Repetitive sequences, for example, can impair synthesis, and self-annealing and hairpin formation increase with prolonged oligos.”

Some technical corrections of the text would be beneficial. In all past of the text the use of terms applicable only for DNA or RNA is mixed and creates some confusion. For example, authors state that "the human cytomegalovirus promoter (CMV) was cloned upstream of 5' UTR and poly(A) tail, the hepatitis delta ribozyme (HDVr) and the simian virus 40 polyadenylation signal downstream of the 3' UTR". Strictly speaking it is impossible as such a construct would contain dsDNA sequence (CMV promoter) followed by ssRNA (5'UTR, polyA tail and HDV ribozyme) and then again dsDNA (SV40 terminator). So, better to be correct and add "sequences corresponding to", "dsDNA copies of" to the description of RNA elements

We thank the reviewer for the advice but would like to state that in scientific language it is common to assume that nucleic acid cloning is based on DNA.

We have corrected the description in the Methods section: “The human cytomegalovirus promoter (CMV) was cloned upstream of the DNA sequence of the viral 5’UTR; herein, the first five nucleotides (ATATT) correspond to the 5’UTR of SARS-CoV. Sequences corresponding to the poly(A) tail (n=35), the hepatitis delta virus ribozyme (HDVr), and the simian virus 40 polyadenylation signal (SV40pA) were cloned immediately downstream of theDNA sequence of the viral 3’UTR.”

For ease of reading and for consistent terminology, we kept the original spelling in the rest of the manuscript.

In description of neutralization assay authors have used temperature 34 C for incubation of virus with antibodies as well as for subsequent incubation of infected cells. Why this temperature was used?

The following sentence was added (Materials and Methods; Cells): “A lower incubation temperature was chosen based on previous studies (V’kovski et al., 2021).”

References

Lett MJ, Otte F, Hauser D, Schön J, Kipfer ET, Hoffmann D, Halwe NJ, Ulrich L, Zhang Y, Cmiljanovic V, Wylezich C, Urda L, Lang C, Beer M, Mittelholzer C, Klimkait T. 2023. Single-cycle SARS-CoV-2 vaccine elicits high protection and sterilizing immunity in hamsters. doi:10.1101/2023.05.17.541127

Liu Y, Zhang X, Liu J, Xia H, Zou J, Muruato AE, Periasamy S, Kurhade C, Plante JA, Bopp NE, Kalveram B, Bukreyev A, Ren P, Wang T, Menachery VD, Plante KS, Xie X, Weaver SC, Shi P-Y. 2022. A live-attenuated SARS-CoV-2 vaccine candidate with accessory protein deletions. Nat Commun 13:4337. doi:10.1038/s41467-022-31930-z

V’kovski P, Gultom M, Kelly JN, Steiner S, Russeil J, Mangeat B, Cora E, Pezoldt J, HolwerdaM, Kratzel A, Laloli L, Wider M, Portmann J, Tran T, Ebert N, Stalder H, Hartmann R, Gardeux V, Alpern D, Deplancke B, Thiel V, Dijkman R. 2021. Disparate temperaturedependent virus–host dynamics for SARS-CoV-2 and SARS-CoV in the human respiratory epithelium. PLoS Biol 19:e3001158. doi:10.1371/journal.pbio.3001158